# Initial elevations in glutamate and dopamine neurotransmission decline with age, as does exploratory behavior, in LRRK2 G2019S knock-in mice

Mattia Volta[1†#], Dayne A Beccano-Kelly[1†¶], Sarah A Paschall[1,2†], Stefano Cataldi[1,2‡], Sarah E MacIsaac[1,2‡], Naila Kuhlmann[1,2,3], Chelsie A Kadgien[1,2,3], Igor Tatarnikov[1], Jesse Fox[1], Jaskaran Khinda[1], Emma Mitchell[1], Sabrina Bergeron[1], Heather Melrose[4], Matthew J Farrer[1§], Austen J Milnerwood[1,3§*]

[1]Centre for Applied Neurogenetics, University of British Columbia, Vancouver, Canada; [2]Graduate Program in Neurosciences, University of British Columbia, Vancouver, Canada; [3]Department of Neurology and Neurosurgery, Montreal Neurological Institute, McGill University, Montreal, Canada; [4]Mayo Clinic, Jacksonville, United States

*For correspondence:
austen.milnerwood@mcgill.ca

[†]These authors contributed equally to this work
[‡]These authors also contributed equally to this work
[§]These authors also contributed equally to this work

Present address: [#]Institute for Biomedicine, Eurac Research, Bolzano, Italy; [¶]Department of Physiology, Anatomy and Genetics, University of Oxford, Oxford, United Kingdom

Competing interests: The authors declare that no competing interests exist.

**Abstract** LRRK2 mutations produce end-stage Parkinson's disease (PD) with reduced nigrostriatal dopamine, whereas, asymptomatic carriers have increased dopamine turnover and altered brain connectivity. LRRK2 pathophysiology remains unclear, but reduced dopamine and mitochondrial abnormalities occur in aged G2019S mutant knock-in (GKI) mice. Conversely, cultured GKI neurons exhibit increased synaptic transmission. We assessed behavior and synaptic glutamate and dopamine function across a range of ages. Young GKI mice exhibit more vertical exploration, elevated glutamate and dopamine transmission, and aberrant D2-receptor responses. These phenomena decline with age, but are stable in littermates. In young GKI mice, dopamine transients are slower, independent of dopamine transporter (DAT), increasing the lifetime of extracellular dopamine. Slowing of dopamine transients is observed with age in littermates, suggesting premature ageing of dopamine synapses in GKI mice. Thus, GKI mice exhibit early, but declining, synaptic and behavioral phenotypes, making them amenable to investigation of early pathophysiological, and later parkinsonian-like, alterations. This model will prove valuable in efforts to develop neuroprotection for PD.
DOI: https://doi.org/10.7554/eLife.28377.001

## Introduction

Parkinson's disease (PD) is clinically diagnosed when patients are presented with characteristic progressive motor symptoms, although post-mortem detection of Lewy pathology and nigral cell loss are currently required for confirmation. A recent study suggests nigral cell death may be as low as 0–10% 1–3 years from diagnosis, whereas dopamine functional markers such as tyrosine hydroxylase (TH) and dopamine transporter (DAT) are profoundly reduced at the earliest points assessed (*Kordower et al., 2013*). The rapid and near complete loss of dopamine functional markers at, or within a few years of, diagnosis argues that ongoing clinical deterioration over several years is due to loss of compensatory mechanisms and/or dysfunction of non-dopaminergic neurons.

Although motor symptoms respond well to current therapy (e.g., dopamine replacement by L-DOPA or deep brain stimulation; DBS), PD is a multisystem disorder with a host of L-DOPA

unresponsive features. All patients suffer a range of non-motor symptoms, many of which appear to precede motor onset by years or decades (*Goldman and Postuma, 2014*; *Chaudhuri and Schapira, 2009*; *Jenner et al., 2013*; *Weintraub et al., 2008a*, *Weintraub et al., 2008b*, Weintraub et al., 2008c*Weintraub et al., 2008c*; *Tolosa et al., 2007*). Cognitive dysfunction is seen in ~40% of newly diagnosed PD cases (*Broeders et al., 2013*; *Pedersen et al., 2013*; *Litvan et al., 2011*) in the form of deficits in attention, executive function, verbal fluency and visuospatial processing, rather than memory per se (although memory is also often impaired) (*Weintraub et al., 2008b*; *Goldman et al., 2015*). This dysexecutive/subcortical syndrome is thought to be due to impaired cortico-striatal basal ganglia processing for action selection (*Goldman and Postuma, 2014*; *Calabresi et al., 2014*; *Gerfen and Surmeier, 2011*). There are no effective symptomatic treatments for many of these non-motor issues, nor are there currently any disease-modifying (neuroprotective) interventions.

While the etiology for most Parkinson patients remains unknown, aside from age, gene mutations contribute the most risk (*Volta et al., 2015a*). Pathogenic mutations in leucine-rich repeat kinase 2 (LRRK2) account for ~1% of all PD cases, of which LRRK2 G2019S is the most frequent; identified in ~30% of cases in some ethnicities (*Ozelius et al., 2006*; *Hulihan et al., 2008*). Affected LRRK2 individuals develop a late-onset, L-DOPA-responsive motor parkinsonism that is clinically and often pathologically indistinguishable from idiopathic PD (*Haugland, 2002*; *Gaig et al., 2014*). Dopamine PET imaging of affected LRRK2 mutation carriers reveals progressive neurochemical alterations similar to those of sporadic PD, namely impaired presynaptic dopamine function (*Adams et al., 2005*; *Nandhagopal et al., 2008*). Further study in LRRK2 families reveals surprising alterations in clinically asymptomatic, otherwise healthy, mutation carriers, including: (i) early increases in dopamine turnover by PET (*Sossi et al., 2010*), (ii) changes in cortical connectivity by resting state MRI and neurochemical changes (*Adams et al., 2005*; *Helmich et al., 2015*; *Vilas et al., 2015*), and (iii) alterations in cognitive tests of executive function (*Thaler et al., 2013*). Advances in our understanding of PD argue cell death and overt motor dysfunction are late occurrences, preceded by dysfunction of dopaminergic and non-dopaminergic systems. In this light, in model systems modelling PD etiology, the underlying pathophysiology and phenotypes should also be expected to be initially subtle, progressive, and include dysfunction of multiple neuronal systems, prior to cell loss.

We engineered the LRRK2 G2019S substitution into the endogenous mouse gene (G2019S knock-in mice; GKI) which produced reductions in basal and pharmacologically evoked nigrostriatal dopamine release in vivo in mice aged >12 months by microdialysis (*Yue et al., 2015*). This Parkinson's-like deficit was not observed at 6 months, and occurred despite a normal compliment of nigral neurons and nigrostriatal dopamine markers (TH). Contrastingly, in cortical neurons cultured from the same GKI mice, we observed increases in glutamatergic and GABAergic synaptic transmission at 21 days in vitro (*Beccano-Kelly et al., 2014*).

To investigate this disparity, we probed dopamine and glutamate release, neuronal morphology, synaptic proteins and behavior in young and aged GKI mice. Young animals exhibit increased exploratory rearing behavior and increases in striatal glutamate and dopamine transmission. As GKI mice age, they exhibit less exploratory rearing and reductions in both glutamate and dopamine transmission. However, extracellular lifetime of single dopamine release events is enhanced in young GKI animals and maintained in aged animals, at which point wild-type littermate values have increased to parity. Several measures demonstrate that the LRRK2 G2019S mutation produces alterations in young adult mice, most of which decline with age, prior to ages where hypodopaminergia, mitochondrial and tau pathology are observed. We provide further evidence that GKI mice provide a valuable model in which to probe early pathophysiological effects of mutant LRRK2 and later classically PD-like deficits. We conclude that understanding the early pathophysiological changes in etiological models may offer the best hope for development of neuroprotection in PD and related diseases.

## Results

### Animal weight and behavioral assessment

Recently, *Longo et al. (2014)* reported that a similar line of homozygous GKI mice weighed significantly less (~15%) than littermates (*Longo et al., 2014*). This conclusion was drawn from longitudinal observations on a cohort of 11 mutants and 9 WT littermates, and separate age-matched

comparisons of 6–8 mice per group. We did not detect genotype-dependent weight alterations in our founding colony at Mayo Clinic, Florida (*Yue et al., 2015*). Here, we found no reductions in weight in comparisons of multiple litters in the UBC colony, over a wide range of ages (*Figure 1A.* total animals WT n = 171, GKI heterozygous n = 276 mice and see *Figure 1—figure supplement 1*. GKI homozygous n = 162). Conversely, while GKI mice were indistinguishable from WT littermates for up to 500 days, a significant genotype-age interaction was detected due to small but significant increases in mean weight of both heterozygous and homozygous GKI animals at 500–600 days (*Figure 1A.*; 2-way ANOVA, genotype $F_{(2,588)}$=1.8, p=0.17, interaction $F_{(12,588)}$=1.7, p=0.049, and see *Figure 1—figure supplement 1*). LRRK2 G2019S parkinsonism is inherited as a Mendelian

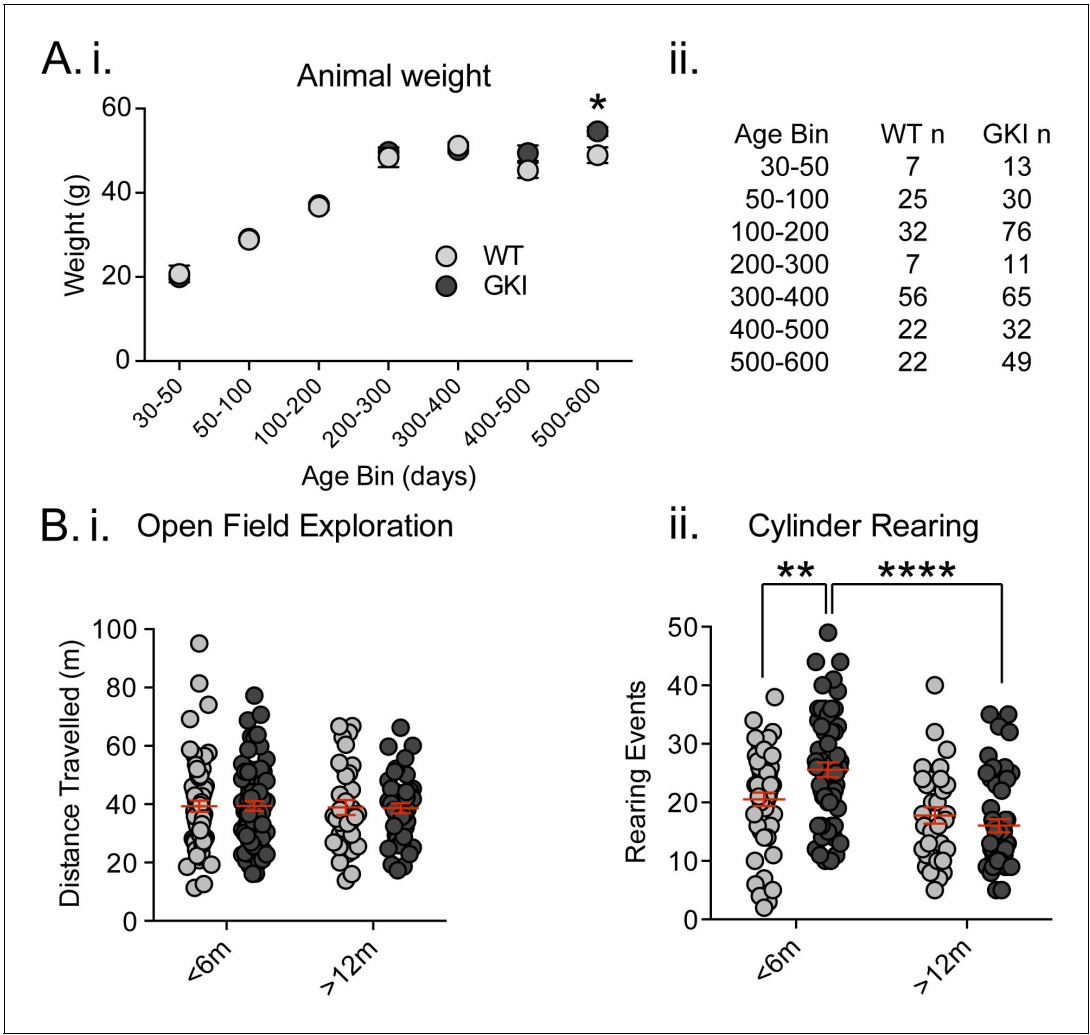

**Figure 1.** Increased exploratory behavior in young GKI mice declines with age. (**A**) Animal body weight over time. There was no main effect of genotype upon mouse weight, but there was a significant interaction between weight and age (see text and *Figure 1—figure supplement 1*). Weights were not significantly different at any age up to 500 days (i; animal n shown in ii); however, in the oldest age group there was a statistically significant increase in GKI mean weight (post-test *p=0.018). (**B**) Spontaneous exploratory behavior in young and old mice. Analysis of open field exploration showed no genotype (p=0.9) or interaction (p=0.9) effects on total distance travelled (i). In the cylinder test for vertical exploration (ii) there was a significant interaction between genotype and age upon rearing events (p=0.008). At < 6 months, GKI mice exhibited a significant 24.8% increase in vertical exploration (post-test **p=0.009), which declined significantly with age (post-test ****p=0.0001). No significant age-related decline was observed in WT littermates (post-test p=0.3).

DOI: https://doi.org/10.7554/eLife.28377.002

The following figure supplement is available for figure 1:

**Figure supplement 1.** Weights of WT, GKI and GKI homozygous animals are similar until advanced ages.

DOI: https://doi.org/10.7554/eLife.28377.003

dominant condition and manifests in heterozygous carriers (*Trinh et al., 2014*; *Kachergus et al., 2005*). Therefore, we focused our effort on comparatively large cohorts of male WT littermate and heterozygous GKI (herein GKI) mice. Based on our previous results (*Yue et al., 2015*) we chose to examine two life stages; early ages when striatal dopamine levels were previously normal by microdialysis (1–6 months;<6 m) and an aged time point when striatal dialysate dopamine levels were reduced (12–18 months;>12), but prior to detection of mitochondrial abnormalities (observed at >22 months [*Yue et al., 2015*]).

*Longo et al. (2014)* found homozygous GKI mice exhibited hyperactivity at all ages tested (*Longo et al., 2014*); although such a result might be impacted by lower body weight in mutants. In agreement, we previously observed an increase (~10%) in open field activity in a small cohort of homozygous GKI mice at 6 (but not 12) months in the founding colony (*Yue et al., 2015*); whereas hyperactivity was not observed at any age in heterozygous mice (*Yue et al., 2015*). Here we tested large cohorts in the open field exploration task and, in agreement with our previous report, found no significant effects (*Figure 1B.i*, WT n = 66 and 34, GKI n = 72 and 44, for <6 and >12 months, respectively; 2-way ANOVA: $F_{(1,212)}$=0.09, 0.002, 0.007 and p=0.8, 0.9 and 0.9 for age, genotype and interaction, respectively). We similarly found no evidence for thigmotaxis at either age point, suggesting a lack of anxiety- or anxiolytic-like phenotypes, which may have altered open field exploration (center path ratios were not significantly different).

Contrastingly, in the cylinder exploration test conducted in a smaller environment that may stimulate mice or relieve some stress of open field testing, we found significant age and age-genotype interaction effects (*Figure 1B.ii*, WT n = 50 and 33, GKI n = 59 and 49, for <6 and >12 months, respectively; 2-way ANOVA: $F_{(1,187)}$=23.6, 7.3, and p=0.0001, 0.008 for age and genotype-age interaction, respectively). Post-test analysis demonstrated a significant ~25% increase in rearing events in GKI mice aged <6 m, relative to WT littermates (p=0.009). This phenomenon was significantly reduced with age in GKI mice (p=0.0001) but not WT animals (p=0.3).

Together, the behavioral data demonstrate that the G2019S mutation (at disease-relevant heterozygous physiological expression levels) results in alterations to behavior in young adult mice, independent of changes to animal weight. Here, in GKI mice, this manifested as early increases in exploratory rearing without changes in open field exploration and thus likely reflects altered motivation for exploration, rather than motor function per se. We conclude, in broad agreement with previous reports (*Yue et al., 2015*; *Longo et al., 2014*), that GKI mice have a propensity to exhibit hyperactivity, but one that is context dependent. Future experiments examining exploration in more complex environments and/or cognitive tasks may prove enlightening at young and aged time points.

## Striatal glutamate transmission

Glutamate release is augmented in networks of cultured GKI cortical neurons, as demonstrated by a > 35% increase in miniature excitatory postsynaptic current (mEPSC) frequency at 3 weeks in vitro (*Beccano-Kelly et al., 2014*). To determine whether similar alterations to cortical/thalamic glutamate release occur in the GKI mouse brain, we conducted whole-cell patch-clamp recording of dorsolateral striatal medium-sized spiny projection neurons (MSNs or SPNs; referred to herein as SPNs) in slices acutely prepared from young and aged GKI and WT littermate mice (*Figure 2*). Intrinsic membrane properties were as predicted for SPNs (*Beccano-Kelly et al., 2015*; *Milnerwood et al., 2013*; *Milnerwood et al., 2010*) and, although there were statistically significant age effects upon membrane capacitance (Cm), membrane resistance (Rm) and decay time constants (Tau m), there were no genotype or genotype-age interaction effects (see *Figure 2—figure supplement 1*; 2-way ANOVA values included). Thus, intrinsic membrane properties of SPNs are not altered by the presence of G2019S LRRK2.

Analysis of spontaneous EPSCs (sEPSCs, *Figure 2A*) revealed a significant main age effect upon event amplitude (*Figure 2A.i*. 2-way ANOVA; $F_{(1,165)}$=3.92, p=0.049, WT n = 40(17) and 31(12), GKI n = 53(20) and 44(15), for 1–3 and >12 months, respectively), but no genotype or genotype-age interaction effects ($F_{(1,165)}$=2.1, 1.8, p=0.15 and 0.18, respectively). Contrastingly, analysis of sEPSC frequencies revealed significant main genotype and age effects (*Figure 2A.ii*. 2-way ANOVA; $F_{(1,165)}$=6.3, 17.3, p=0.013, 0.0001, respectively), and a significant interaction between genotype and age ($F_{(1,165)}$=7.8, p=0.006). Post-tests demonstrated a significant ~47% increase in sEPSC frequency in GKI SPNs aged 1–3 months, relative to WT littermates (p=0.0004). This phenomenon was

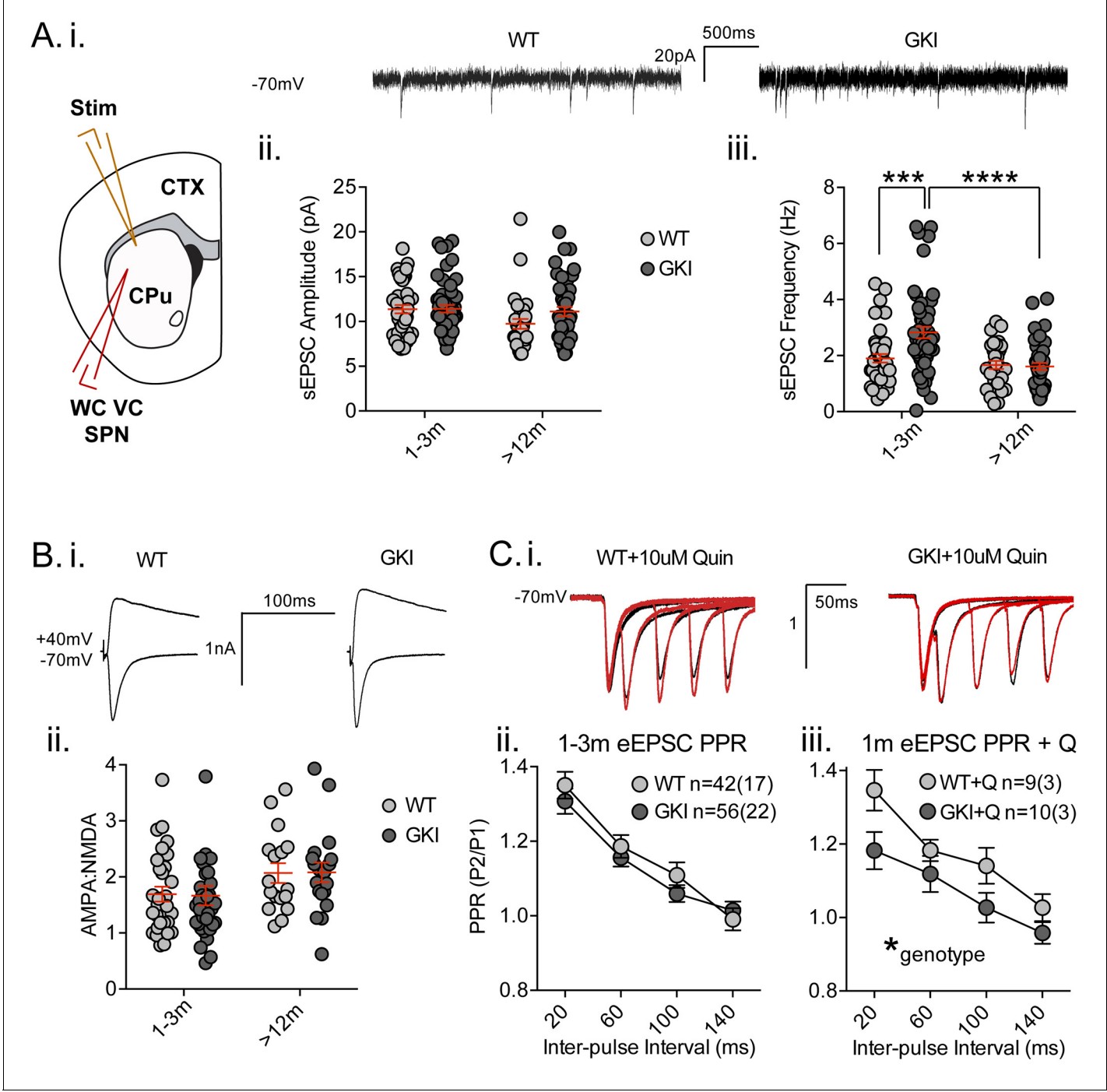

**Figure 2.** Increased glutamate transmission onto young GKI striatal projection neurons declines with age. (A) Whole-cell voltage-clamp (WC VC) recording in striatal medium-sized spiny projection neurons (SPNs) of the dorsolateral striatum of acutely prepared brain slices. Recording electrodes targeted visually identified SPNs in the dorsolateral caudate putamen (CPu) of the striatum and stimulating electrodes were placed intrastriatally ~200 um dorsal of the recording site toward M1/S1 cortex (CTX) (i). Example traces of spontaneous activity (spontaneous excitatory post-synaptic currents: sEPSCs) in slices from 1m-old WT (left) and GKI (right) mice. There was a significant main effect of age upon mean sEPSC amplitude (ii, p=0.049), but no main genotype (p=0.15) or age-genotype interaction (p=0.18). In contrast, there were significant main age (0.0001) and genotype (p=0.013) effects and an age-genotype interaction effect (p=0.006) upon the frequency of sEPSCs. At 1–3 months, GKI SPNs exhibited a significant 48.4% increase in mean event frequency (post-test **p=0.0004), which significantly declined (by 57%) with age (post-test ****p=0.0001). In contrast, the 13% decline with age in WT SPNs was not significant (post-test p=0.29). (B) Representative traces of intrastriatally-evoked AMPA- and NMDA-receptor mediated EPSCs (eEPSCs) recorded at Vh −70 mV and + 40 mV, respectively (i). There was a significant main effect of age (ii; p=0.041) upon the ratio of AMPA to NMDA eEPSC amplitudes, but no main genotype (p=0.96) or age-genotype interaction effect (p=0.92). (C) Representative peak-normalized traces of evoked

*Figure 2 continued on next page*

*Figure 2 continued*

EPSCs (eEPSCs) in SPNs, resulting from intrastriatal paired-pulse stimuli (20–140 ms inter-pulse intervals; IPIs) prior to (i. black trace) and in the presence of 10 uM quinpirole (red trace; slices from 1 month WT and GKI slices). There were no significant genotype or genotype-IPI interaction effects upon paired-pulse ratios in pooled experiments from animals aged 1–3 months in standard conditions (ii). In the presence of the D2 agonist quinpirole (i) there was a significant main genotype effect (*p=0.042) in slices from 1 m old mice (iii).

DOI: https://doi.org/10.7554/eLife.28377.004

The following figure supplements are available for figure 2:

**Figure supplement 1.** There are no genotype-dependent alterations to SPN intrinsic membrane properties.
DOI: https://doi.org/10.7554/eLife.28377.005

**Figure supplement 2.** GKI SPN sEPSC frequency is similarly increased in 1 and 3 month slices, and similar to WT in 12 and 18 month slices.
DOI: https://doi.org/10.7554/eLife.28377.006

**Figure supplement 3.** Spontaneous EPSCs recorded in dorsolateral striatum in the coronal slice preparation are TTX insensitive.
DOI: https://doi.org/10.7554/eLife.28377.007

**Figure supplement 4.** There are no differences in the number of postsynaptic specializations, or presynaptic excitatory nerve terminal markers in the striatum of young or aged GKI mice.
DOI: https://doi.org/10.7554/eLife.28377.008

**Figure supplement 5.** Paired-pulse facilitation profiles in the striatum of aged GKI mice are similar and GKI mutant PPRs are insensitive to dopamine agonism and antagonism.
DOI: https://doi.org/10.7554/eLife.28377.009

**Figure supplement 6.** Spontaneous EPSCs recorded in GKI dorsolateral striatum SNPs are remoxipride insensitive.
DOI: https://doi.org/10.7554/eLife.28377.010

**Figure supplement 7.** There were no differences in GKI sEPSC frequency between D1- and D2-dopamine receptor expressing SPNs, and no consistent effect of quinpirole upon eEPSC peak or PPR in either cell type.
DOI: https://doi.org/10.7554/eLife.28377.011

---

significantly reduced with age in GKI mice (p=0.0001), but not WT animals (p=0.6). To more discretely interrogate the age-dependency of this large increase in sEPSC event frequency, comparisons were made between GKI and WT data sets in approximate 1, 3, 12 and 18 month groupings (*Figure 2—figure supplement 2*, RM-ANOVA values included); inter-event interval cumulative probabilities demonstrated that frequency is higher in GKI SPNs at 1 and 3, but not 12 or 18 months of age.

A recent report by *Matikainen-Ankney et al. (2016)* also found a similar increase in SPN event frequency in slices from similar GKI mice (aged < 1 month) (*Matikainen-Ankney et al., 2016*), which was concluded to be predominantly due to action potential-mediated transmission (only a strong trend toward increased frequency was observed in the presence TTX). We have previously demonstrated that sEPSCs in our coronal slice preparation are TTX-insensitive, and thus predominantly miniature EPSCs (mEPSCs) (*Milnerwood et al., 2010*); however, the possibility remains that an increase in basal GKI event frequency is due to anomalous preservation of action potential-dependent release. We found that GKI event amplitudes and frequencies were insensitive to TTX application (*Figure 2—figure supplement 3*), with only a weak trend toward increased (not reduced) frequency following blockade of action potential firing. We conclude that sEPSCs in GKI slices are predominantly, if not entirely, independent of action potentials and therefore represent miniature release.

A simple explanation for increased glutamatergic event frequency is an increase in the number of synapses (with equal release probability); however, we found no significant genotype effect upon postsynaptic dendritic spine-like structures (quantified from Golgi impregnation), or immunohistochemical staining of presynaptic vesicular glutamate transporters 1 and 2 (VGluT1, VGluT2; *Figure 2—figure supplement 4*). Thus, similarly to glutamatergic mEPSCs in cortical cultures at 3 weeks in vitro (*Beccano-Kelly et al., 2014*), GKI SPNs in slices from young mice exhibited no difference in sEPSC event amplitude but significant increases in event frequency in the absence of changes to synapse numbers. These specific early increases in glutamate release frequency (as opposed to event amplitude) are not maintained in SPNs of slices from older animals; in agreement with *Matikainen-Ankney et al. (2016)* who also report no changes to synapse density (*Matikainen-Ankney et al., 2016*).

Synchronous glutamate release was also investigated by intrastriatal stimulation-evoked excitatory postsynaptic currents (eEPSCs; *Figure 2B and C*). While raw amplitudes of eEPSCs are difficult

to interpret (due to potential differences in stimulating electrode placement), within-cell measures such as the ratio of AMPA-receptor mediated (Vh −70 mV) to NMDAR-mediated (Vh + 40 mV) current amplitudes (AMPA:NMDA ratios) and paired-pulse ratios (PPRs) are widely used to estimate postsynaptic responsiveness and presynaptic release probability, respectively. Although there was a significant effect of age, we found no significant genotype effect upon AMPA:NMDA ratios of GKI and WT eEPSCs (*Figure 2B*, 2-way ANOVA; $F_{(1,106)}$=4.3, 0.002, p=0.041, 0.97 for age and genotype, respectively). In the face of elevated glutamate event frequency, a lack of change in the ratio (AMPA responsiveness and synapse numbers) is often interpreted as evidence for alterations at the presynaptic terminal.

Paired-pulse ratios (often seen as indicative of presynaptic release probability) were not significantly altered by genotype in slices from young (*Figure 2C.i and ii*, 2-way RM-ANOVA; $F_{(1,96)}$=0.51, p=0.47) or aged mice (*Figure 2—figure supplement 5*, 2-way RM-ANOVA; $F_{(1,96)}$=0.51, p=0.47). Several reports demonstrate a disconnection between alterations in miniature event frequency and predicted changes to PPRs (*Milnerwood et al., 2013*; *Cepeda et al., 2008*; *Sara et al., 2005*; *Ramirez et al., 2012*; *Crawford and Kavalali, 2015*; *Milnerwood and Raymond, 2007*; *Parisiadou et al., 2014*; *Mei et al., 2016*; *Akopian and Walsh, 2007*) and recent evidence suggests the mechanisms for miniature and synchronous release may be distinct (reviewed in [*Crawford and Kavalali, 2015*]). Therefore, specific alterations to spontaneous release may entirely underlie the increase observed in GKI slices. Alternatively, paired-pulse facilitation ratios are reduced (and release probability is increased) in thalamostriatal, relative to corticostriatal, terminals (*Ding et al., 2008*; *Sciamanna et al., 2015*). Thus, altered contributions of one or both pathways might account for observed differences in young (and potentially masked differences in old) GKI slices. Ongoing experiments selectively probing spontaneous and evoked miniature release in both pathways may prove informative.

Upon intrastriatal stimulation, glutamatergic terminals (and, by inference, PPRs) are subject to modulation by the simultaneous release of dopamine (*Bamford et al., 2004*), among other transmitters. In slices from LRRK2 overexpressing mice, we found PPR alterations were dependent upon the activity of presynaptic D2 receptors (*Beccano-Kelly et al., 2015*). Here, in the presence of the D2 receptor agonist quinpirole (Q; 10 uM, bath applied) we found a significant main effect of genotype upon PPRs; GKI currents were reduced relative to WT controls (*Figure 2C.iii*, 2-way RM-ANOVA; $F_{(1,17)}$=4.8, p=0.042), indicative of a relatively higher initial release probability at GKI synapses, when D2 receptors were stimulated in both genotypes. This difference is likely due to D2 agonism having increased WT PPRs in this subset of cells (n = 9 (3); 2-way RM-ANOVA treatment $F_{(1,32)}$=7.5, p=0.01 before vs. after quinpirole), by presynaptic D2 negative tuning of initial release probability (and resultantly increased PPR) (*Beccano-Kelly et al., 2015*; *Bamford et al., 2004*). Conversely, quinpirole had no effect on PPR in GKI (*Figure 2—figure supplement 5*), suggesting that D2 negative tuning upon presynaptic terminals is dysfunctional, or saturated, at GKI synapses. The D2 antagonist remoxipride (10 uM, bath applied) also had no significant effect upon GKI PPRs (*Figure 2—figure supplement 5*); suggesting that acute D2 activation on glutamatergic terminals is dysfunctional, not saturated. This loss of dopamine-dependent negative regulation may contribute to basal increases in GKI event frequency at this young age, if intrinsic differences in glutamate release are beyond modulation by this form of negative regulation (*Bamford et al., 2004*).

SPNs are equally subdivided by postsynaptic D1 or D2 dopamine receptor expression and form the direct and indirect striatal output pathways, respectively. Paired-pulse ratios were initially shown to be elevated in direct pathway SPNs (*Kreitzer and Malenka, 2007*), and although subsequent reports found no difference in PPRs (*Cepeda et al., 2008*; *Ding et al., 2008*), miniature EPSC frequencies appear consistently higher in D2 indirect pathway SPNs (*Cepeda et al., 2008*; *Kreitzer and Malenka, 2007*). To visually target D1 (red) and D2 expressing striatal SPNs, a subset of GKI mice were crossed with line 6 *Drd1a*-tdTomato ± mice (*Ade et al., 2011*) and cell filling during recording and post hoc staining were conducted to verify visual calls. We found no cell-type or age-effect on sEPSC event frequencies of D1 or D2 SPNs in slices from GKI mice (*Figure 2—figure supplement 7*, 2-way ANOVA; $F_{(1,19)}$=1.23, 0.56, p=0.21, 0.46 for cell type and age, respectively). We also found that quinpirole application did not have a consistent effect upon eEPSC amplitude or paired-pulse ratios in either D1 or D2 SPNs from GKI mice (*Figure 2—figure supplement 7*). Similarly, remoxipride had no effect upon event amplitude or frequency in GKI SPNs (*Figure 2—figure supplement 6*).

Together the data suggest that presynaptic glutamate transmission onto SPNs of GKI mice is augmented (due to altered release, not changes in synapse number) at an early age when presynaptic D2 tuning is also dysfunctional. These initial levels of activity decline over the animals' lifetime, unlike WT SPNs.

## Striatal dopamine release

We previously found a significant reduction in basal extracellular dopamine levels in GKI mice aged 12, but not 6 months (*Yue et al., 2015*). As no differences in the total amount of dopamine were detected, reduced extracellular dopamine in old animals was interpreted as a latent impairment of dopamine release. In light of age-dependent alterations to glutamate transmission, we resolved to test dopamine release directly by fast-scan cyclic voltammetry in the dorsolateral striatum young and aged mice (*Figure 3*). Single pulses evoking dopamine release by increasing stimulus intensities (50 - ~450 uA, 0.01 Hz) revealed similar peak currents and calibrated extracellular dopamine peak concentrations in slices from young GKI and WT animals; there was no effect of genotype (*Figure 3A.i and ii*. 2-way ANOVA; $F_{(1,140)}=0.001$, p=0.97). Single pulse evoked dopamine release had declined by approximately 50% in aged animals, (*Figure 3A.ii*), but again there was no effect of genotype (2-way ANOVA; $F_{(1,141)}=1.79$, p=0.18). The data suggest there is no basal impairment to stimulated dopamine release in aged GKI mice, despite reduced extracellular dopamine in vivo. To assess the stability of dopamine transmission, single stimuli were set to evoke ~70% maximum release every 2 min (0.01 Hz; *Figure 3A.ii*). Over 10 min, in slices from young animals, we found a significant main effect of genotype (2-way RM-ANOVA; $F_{(1,53)}=8.5$, p=0.005), and a significant interaction between genotype and pulse number ($F_{(4,212)}=4.0$, p=0.003). Where WT amplitudes declined, responses in young GKI slices increased such that there was significantly more dopamine released (*Figure 3A.ii*). Conversely, in slices from aged mice, neither genotype exhibited a relative increase or decrease in dopamine release over the same timeframe (*Figure 3A.ii*; 2-way RM-ANOVA; $F_{(1,61)}=0.65$, p=0.4). The data suggest that the GKI mutation results in alterations to the regulation of dopamine release in young animals, rather than disturbing the mechanics of single stimuli-evoked release. Furthermore, this alteration is no longer present at 12 months.

In slice experiments, striatal dopamine release is curtailed over short time periods (<5 s) by D2 autoreceptors on nigral terminals (*Phillips et al., 2002*; *Rice et al., 2011*), evident as paired-pulse depression. Paired-pulses (*Figure 3A.i and iii*) evoked predicted depression of the second pulse in WT and GKI slices from young (~70%) and aged animals (~80%); there was a significant effect of age, but not genotype on paired-pulse ratios (*Figure 3A.iii*. 2-way ANOVA; $F_{(1,77)}=37.9$, 0.142 p=0.0001 and 0.7, respectively; post-test 3 month vs. > 12 month p=0.0002 and 0.0002 for WT and GKI, respectively). The data demonstrate that D2 autoreceptor signaling and short-term negative regulation of dopamine release is not altered by the GKI mutation. To investigate chronic D2 activation, we bath applied a low concentration of the D2 agonist quinpirole (50 nM, *Figure 3B.i*). In the initial presence of quinpirole (0 min; 10 min after switching perfusate), response peaks of slices from mice aged 3 months were reduced by ~20% in both genotypes. However, upon continued stimulation in the presence of quinpirole, the depressive effect became more pronounced in GKI than WT slices as highlighted by a significant interaction between genotype and time (*Figure 3B.i*. 2-way RM-ANOVA; $F_{(4,108)}=2.6$, p=0.039). By 18 min post-application, WT values were reduced to $64.7 \pm 5.4\%$, whereas GKI were significantly more sensitive, being reduced to $46.75 \pm 3.29\%$ (post-test p=0.049). In slices prepared from mice aged >12 months, no such genotype interaction was observed (*Figure 3B.i*). The data demonstrate that early hypersensitivity of dopamine release to chronic D2 activation in GKI slices is not present in slices from older animals.

Striatal dopamine transmission is tightly regulated by dopamine transporter (DAT)-mediated reuptake; DAT levels, location and activity dictate the sphere of influence and, dominating over diffusion, are the major determinant of released dopamine's extracellular lifetime in the striatum (*Rice et al., 2011*; *Jones et al., 1998*; *Gainetdinov, 2008*). There was a significant main effect of genotype (*Figure 3B.ii*. No Drug; 2-way ANOVA; $F_{(1,81)}=9.73$, p=0.003) upon single response decay times (Tau); decays were significantly slower in slices from GKI mice aged 3 months than WT controls (post-test p=0.01), but similar in older slices. Slower DA transients might be expected to result from reduced DAT levels or activity, but we found no genotype, age or interaction effects upon DAT (or TH) staining (*Figure 3—figure supplement 1A*. ANOVA values included). To further probe DAT we also biochemically assessed protein levels and found a significant main effect of genotype, due to

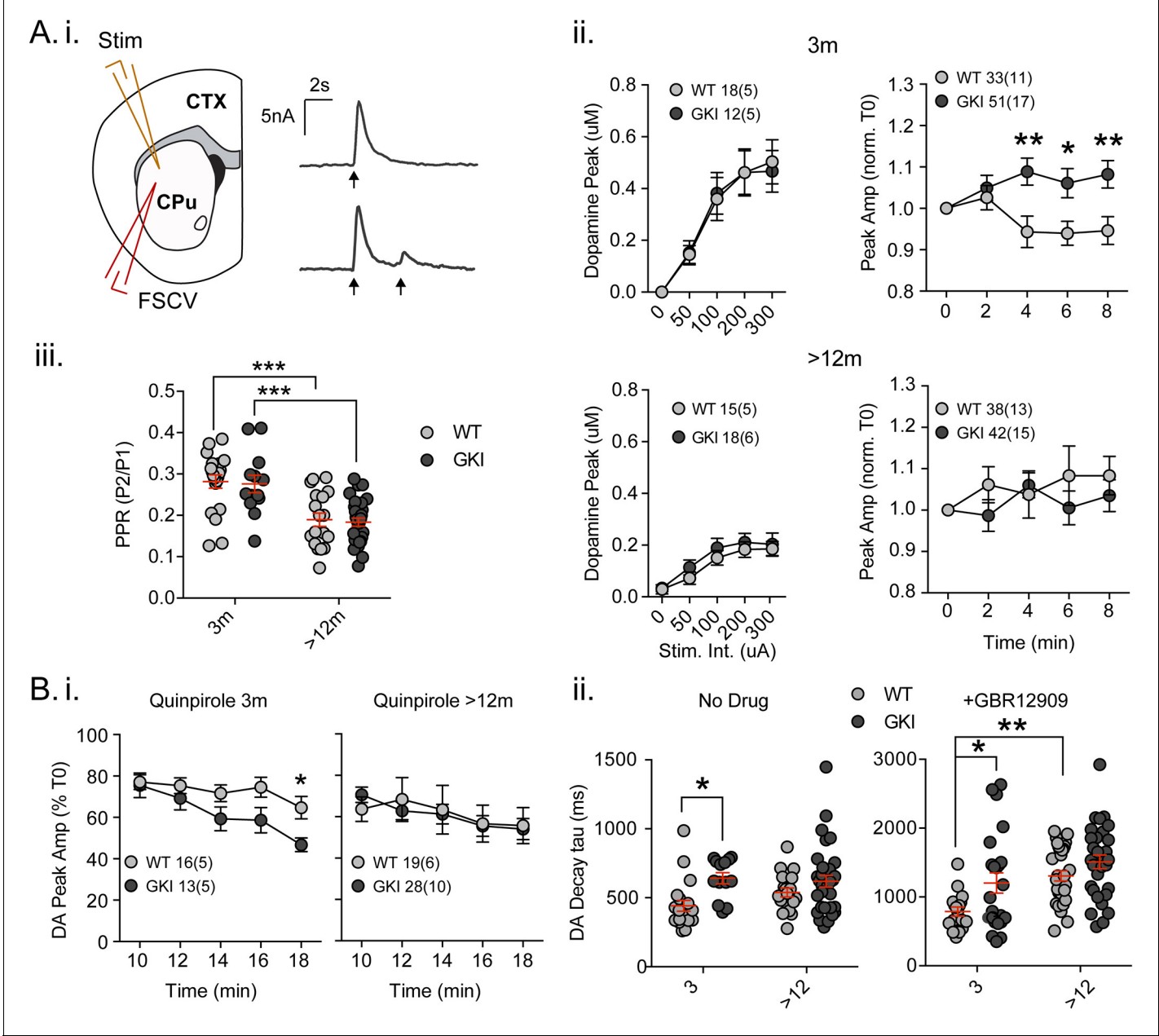

**Figure 3.** Increased nigrostriatal dopamine transmission in young GKI slices is lost with age, whereas early increases in dopamine extracellular lifetime are maintained in old GKI slices, and matched by age-related increases in WT slices. Fast-scan cyclic voltammetry (FSCV) was conducted in the dorsolateral striatum of acute slices prepared from WT and GKI mice. (**A**) Dopamine release and reuptake transients were evoked by single or paired (4 s IPI) local stimuli (i). There was no significant genotype effect (see text) on peak dopamine release over a range of stimulus intensities in slices from 3 month or > 12 month old mice (ii). Upon repeated stimulation, there were significant (post-test *p=0.05, **p=0.01), elevations in dopamine peak release in GKI mice, relative to WT controls at 3 months (ii). In WT and GKI slices from > 12 month mice, peak dopamine release was more variable, there was no modulation induced by repeated stimulation, as in younger slices, nor were there genotype differences. There was a significant age-dependent increase in D2 autoreceptor mediated paired-pulse depression (iii; 3 month n = 20(5) and 13(5),>12 month n = 18(6) and 30(11) for WT and GKI, respectively, post-test ***p=0.001), reflected by reduced paired-pule ratios (PPR), but there was no main effect of genotype (p=0.7) or genotype-age interaction (p=0.99). (**B**) The D2 agonist quinpirole (10 uM) equally suppressed peak evoked dopamine release in WT and GKI slices from 3 month mice after a 10 min wash-in; however, continued exposure revealed a trend toward a main effect of genotype (p=0.07) on repeated release, and a significant genotype-time interaction (p=0.039, post-test *p=0.049 at 18 min post application). This effect was not observed in > 12 month slices (i). Evoked dopamine transient decay time revealed a significant main genotype effect (ii. No drug, 3 month n = 20(5) and 13(5),>12 month n = 19(6) and 33(11) for WT and GKI, respectively p=0.0025) due to significantly longer decay times (slower transients) in 3 month GKI slices (post-test *p=0.027). In > 12 month slices, WT decay times had increased to a value similar to 3 month GKI (ii. No drug). The presence of the presynaptic dopamine transporter (DAT)

*Figure 3 continued on next page*

*Figure 3 continued*

blocker GBR (10 uM, ii. + GBR12909) increased all transient decay times, as expected. There were significant main effects of age and genotype (3 month n = 17(6) and 23(8),>12 month n = 31(9) and 30(11), p=0.0002 and 0.0043 for WT and GKI, respectively) upon transient lifetime; in the absence of dopamine re-uptake, 3 GKI transients were still significantly slower than WT controls (post-test *p=0.014), and significant age-dependent increases in dopamine decay times were observed in WT slices (post-test *p=0.0024).

DOI: https://doi.org/10.7554/eLife.28377.012

The following figure supplement is available for figure 3:

**Figure supplement 1.** Early alterations in GKI dopamine release are not associated with reductions in nigrostriatal dopamine markers and persist despite genotype-dependent increases in DAT protein levels by western blot.

DOI: https://doi.org/10.7554/eLife.28377.013

increased levels DAT protein (relative to GAPDH) in the brains of GKI mice (*Figure 3—figure supplement 1B*. ANOVA values included).

Although increased levels of DAT protein suggests slower transients are not due to a paucity of DAT, it is possible that slowing of DA transients in GKI mice is due to a functional impairment in DAT clearance after release. Therefore, in a separate set of experiments we evoked release in a high concentration of the DAT blocker GBR12909 (1 uM, $IC_{50}$ = 6.63 nM; *Figure 3B.ii*). In the absence of DAT re-uptake, as expected, WT decays were approximately twice as long (No Drug 442.1 ± 38.9 ms,+GBR12909 788.2±ms); however, there remained significant main genotype and age effects (*Figure 3B.ii* +GBR12909; 2-way ANOVA; $F_{(1,97)}$=8.55 and 15.35, p=0.004 and 0.0002, respectively). Consistent with results in the presence of DAT reuptake, decays were significantly longer in slices from GKI mice than WT controls aged 3 months when DAT was blocked (post-test p=0.014). Decays in WT animals slowed with age (post-test p=0.002) to an extent similar to both young and old GKI decay times.

The data demonstrate augmented dopamine transmission in young GKI mice, revealed by repeated stimuli and extracellular lifetime of individual pulses, which is not due to a deficit in DAT-dependent clearance. Moreover, an age-dependent increase in stimulated striatal dopamine signalling (increased extracellular lifetime) appears to occur in GKI mice at an earlier time point. As stimulated release is not impaired in older GKI slices, we conclude that the latent reduction in extracellular dopamine levels observed at 12 (but not 6) months by microdialysis (*Yue et al., 2015*) is a result of changes to the endogenous regulation of nigrostriatal dopamine release, rather than impaired vesicular release per se. Interestingly, in agreement with our data here, another recent report in similar knock-in mice by Longo and colleagues (*Longo et al., 2017*) found that DAT levels and activity are increased in GKI brains from animals aged 12 (but not 3) months. In this light, synaptic DA release seen here may appear similar to WT in aged GKI slices (or even reduced by microdialysis [*Yue et al., 2015*]) due to increased clearance through DAT, masking a persisting increase in DA release.

## Summary and conclusions

Familial and idiopathic PD are now accepted as complex motor and non-motor syndromes resulting from dysfunction to multiple neurotransmitter systems and cell types throughout the peripheral and central nervous system (*Volta et al., 2015a*; *Kalia and Lang, 2015*). Although some treatments are effective for motor dysfunction, none slow or prevent the neurodegenerative process. Many non-motor symptoms antedate motor manifestations (and clinical diagnosis) by years to decades; demonstrating protracted pathological processes. In this light, investigations of PD etiology in model systems, especially short-lived rodents, should expect initially subtle, potentially progressive, dysfunction of multiple neuronal systems, prior to (or without) overt motor dysfunction and cell loss (*Volta et al., 2015a*; *Kalia and Lang, 2015*).

GKI mice harbor a single point mutation, which confers the greatest genetic risk for PD in humans. We find that spontaneous exploration is significantly elevated in certain tasks in young mice, in broad agreement with work by other labs in similar animals (*Longo et al., 2014*). This exploratory hyperactivity is seemingly context dependent, observed in cylinder exploration but not evident in the open field; as such we predict this reflects changes in motivation, rather than motor function *per se*. We expect ongoing investigations of these animals in more complex environments and tasks will uncover other phenotypes. We previously observed that presynaptic glutamatergic

release is elevated in cortical cultures from GKI mice (*Beccano-Kelly et al., 2014*), suggesting altered synapse development and/or mature function. Here we find that similar increases are observed in glutamate release onto striatal neurons in brain slices from young GKI animals, without changes in synapse number. This increase is not preserved as animals age, with GKI mice exhibiting a pronounced down-regulation of spontaneous glutamate release. A recent independent study of similar GKI animals drew similar conclusions (*Matikainen-Ankney et al., 2016*), and the weight of evidence demonstrates early alterations to synaptic function produced by endogenous G2019S mutations.

Our previous study revealed reduced extracellular dopamine levels in the striata of aged (12 months), but not young (6 month) GKI mice (*Yue et al., 2015*). Our analyses of dopamine transmission here demonstrates that reductions in vivo are unlikely to be due to impaired release or DAT reuptake, and are likely a consequence of altered regulation of dopamine release and/or nigral burst firing patterns in vivo. In young GKI mouse slices, dopamine release was elevated upon repeated stimulation, and the extracellular lifetime of dopamine was consistently augmented. This suggests that, similar to glutamate synapses, dopamine transmission is elevated in young animals by the G2019S mutation, an elevation which is lost with ageing. Recent brain imaging studies demonstrate that clinically manifest LRRK2 mutation carriers develop deterioration of the dopamine and serotonin systems, similarly to sporadic PD patients (*Sossi et al., 2010*; *Wile et al., 2017*); however, non-manifest mutation carriers exhibit increased striatal dopamine turnover (*Sossi et al., 2010*) and early increases in serotonin transporter binding (*Wile et al., 2017*). We conclude that the GKI mouse is a valuable model in which to probe the etiology and early pathophysiology of LRRK2, and potentially sporadic, parkinsonism. Interventions against such pathophysiological processes in these models may provide the functional neuroprotection, so desperately lacking for Parkinson's disease and related disorders.

# Materials and methods

## G2019S knock-in mice and behavioral testing

C57Bl/6J wild-type (WT) and *Lrrk2* G2019S knock-in heterozygous (GKI) mice (*Yue et al., 2015*; *Beccano-Kelly et al., 2014*) were maintained according to the Canadian Council on Animal Care regulations. To avoid confounds of oestrus cycle upon behavior and neural connectivity, only male animals were used in this work. Mice undergoing surgery were weighed at the age of use, and all other mice in the colony were weighed at a single time point to produce an age vs. weight plot. Separate cohorts of adult animals were tested (once only) at 1–6 and 12–18 months of age. After familiarization to handling over three days with the operator, mice underwent the following behavioral paradigms and videos were analyzed post-hoc using ANY-maze (Stoelting) behavioral tracking software, as previously (*Beccano-Kelly et al., 2015*; *Volta et al., 2015b*). *Open field (OF) test*: mice explored an arena (48 cm x 48 cm) for 15 min. *Cylinder test*: mice were placed in a 1 l beaker and video recorded for 5 min. The number of rearings and forelimb wall contacts were scored manually offline. All testing and analysis was performed experimenter blind.

## Electrophysiology

Whole-cell patch clamp recording was conducted in striatal projection neurons (SPNs) in 300 µm thick coronal slices from 1 to 18 month-old male mice, as in (*Beccano-Kelly et al., 2015*; *Milnerwood et al., 2010*). To help preserve cell viability, slices from > 12 month old animals were pre-incubated in recovery solution containing (in mM: 93 NMDG, 93 HCl, 2.5 KCl, 1.2 NaH2PO4, 30 NaHCO3, 20 HEPES, 25 Glucose, 5 Sodium Ascorbate, 2 Thiourea, 3 Sodium Pyruvate, 10 MgSO4.7H2O, 0.5 CaCl2.2H2O, pH 7.3–7.4, 300–310 mOsm) for 15 min at 34°C prior to transfer to a holding chamber. Slices were held and perfused at RT with artificial cerebrospinal fluid (ACSF) containing (in mM): 125 NaCl, 2.5 KCl, 25 NaHCO$_3$, 1.25 NaH$_2$PO$_4$, 2 MgCl$_2$, 2 CaCl$_2$, 10 glucose, pH 7.3–7.4, 300–310 mOsm). Cells were visualized by IR-DIC on an Olympus BX51 microscope (20x + 4 x magnifier) and SPNs visually identified by somatic size (8–20 µm), morphology and location within the dorsolateral striatum, 50–150 µm beneath the slice surface. Data were acquired by Multiclamp 700B amplifier digitized at 10 kHz, filtered at 2 kHz and analyzed in Clampfit10 (Molecular Devices). Pipette resistance (Rp) was 5–8 MΩ when filled with (in mM): 130 Cs methanesulfonate, 5 CsCl, 4

NaCl, 1 MgCl$_2$, 5 EGTA, 10 HEPES, 5 QX-314, 0.5 GTP, 10 Na$_2$-phosphocreatine, and 5 MgATP, 0.1 spermine, pH 7.2, 290 mOsm. Tolerance for series resistance (Rs) was < 25 MΩ and uncompensated; ΔRs tolerance was < 10%. Events were analyzed blind using Clampfit10 (threshold 5 pA), and checked by eye so only monophasic events were included for amplitude and decay kinetics, whereas all others were suppressed but included in frequency counts. eEPSCs were evoked by 40μs pulses ranging in intensity between 20-550μA delivered through a glass recording micropipette placed 200–250 μm dorsal of the recording site. Stimulus intensities were set to produce ~60–70% max amplitude response for paired-pulses (20–140 ms inter-pulse intervals) and AMPA:NMDA eEPSCs (AMPA-mediated at Vh −70 mV, NMDAR-mediated Vh + 40 mV, 40 ms delay). Responses were evoked at 0.066 Hz. Data are presented as mean ± SEM where n is cells from a minimum of 3 animals (animal n in brackets). Quinpirole ((-)-quinpirole hydrochloride, 10 uM) and remoxipride ((S)-(-)−3-Bromo-N-[(1-ethyl-2-pyrrolidinylmethyl]2,6-dimethoxybenzamide hydrochloride, 10 uM), purchased from Tocris Biosciences, and tetrodotoxin (TTX; 1 uM, Alomone Labs) were bath applied in the perfusate. To investigate potential cell-specific effects, a subset of GKI mice were crossed with Drd1a-(td) tomato mice (Jax: B6.Cg-Tg(Drd1a-tdTomato)6Calak/J) (*Ade et al., 2011*) to target visually identified D1R (+ve) and D2R (-ve) expressing SPNs; cells were also filled with either biocytin (~0.1%) or Lucifer yellow (~0.02%) in the internal solution and post hoc immunohistochemically stained for substance-P to further identify D1R-expressing or D2R-expressing SPNs (described below).

## Fast-scan cyclic voltammetry

300 um coronal sections were prepared as above, allowed to recover for ≥1 hr, then single sections transferred to the recording chamber (23–25°C). Stimuli were delivered by nickel-chromium wire bipolar electrodes (made in house) placed in the dorsolateral striatum, optically isolated (A365, World Precision Instruments, FL, USA) and controlled/sequenced with ClampEx software. Flow rate was 1–2 ml/min. Voltammetric responses were recorded, standardized and analyzed with an Invilog In Vivo Voltammetry system and software components (Invilog Research Ltd., Kuopio, Finland). Carbon fiber electrodes (diameter: 32 μm, length: 30 μm, sensitivity: 21-40nA/μM) were purchased prefabricated by Invilog and placed within 100–200 um of the stimulator. Triangular waveforms (ramp from −400 mV to 1200 mV to −400 mV, 10 ms duration at 10 Hz) were used to detect the oxidation and redox peak for DA between 700 and 800 mV (at 3.2 and 3.5 ms). Input/Output paradigm consisted of increasingly intense single pulse stimuli (100-700 μA, delivered every 2 minutes / 0.01 Hz) to determine ~70% of the maximum response used for the rest of the experiment. Five single pulses were delivered at 0.01 Hz to assess baseline stability, followed by a single paired-pulse stimulus (4 s IPI). During drug wash in, single pulse stimulations were continued at 0.01 Hz for ten minutes. A repeat of the five single simulations and paired-pulses were then recorded in the drug condition. At the end of each day, a three-point calibration of each carbon fiber electrode was conducted (final concentrations 0 μM, 0.5 μM, 1.0 μM DA in ACSF). All drugs were bath applied in the perfusate; quinpirole ((-)-quinpirole hydrochloride, Tocris Biosciences, Bristol, UK, 50 nM, previously assayed to reduce control dopamine transients by < 50%) and GBR-12909 (1-[2-[Bis-(4-fluorophenyl)methoxy]-ethyl]−4-[3-phenylpropyl]piperazine dihydrochloride; Tocris Biosciences, 1 μM) were employed to assay presynaptic D2R agonism and DAT-dependent DA reuptake, respectively. For subsequent experiments after GBR was applied, to avoid potential confounds of residual GBR in the perfusate, data were not used for measurement of basal response peak amplitudes, paired pulses or response decay kinetics.

## Immunohistochemistry

For a subset of whole-cell recordings, intracellular solution contained biocytin (~0.1%) or Lucifer yellow (~0.02%). Slices containing filled cells were fixed (4% paraformaldehyde in 4% sucrose PBS, 4°C overnight), as previously (*Beccano-Kelly et al., 2015*), washed (PBS), permeabilized (0.3% H$_2$O$_2$ in 100% methanol 30 min), rinsed (3x PBST, 0.1%) and blocked (3% non-fat milk and 10% NGS for 1 hr) prior to incubation with anti-Substance P (SP) antibody (24 hr 4°C, AB1566; Millipore; 1:100), then Alexafluor-488 secondary antibody (2 hr RT Lifetech; 1:200) and Cy3-conjugated streptavidin (for biocytin filled sections) prior to slide mounting. Sections were imaged by confocal laser-scanning microscopy (Olympus Fluoview FV1000). Mean somatic intensity of SP immunolabeling was measured from flattened images produced by z-stack max projections of ~6 (1um z-step) sections

containing the cell body (~5–7 microns of tissue depth). Intensity measures (488 nm) were produced using somatic ROIs constructed with Cy3 or Lucifer yellow channel and normalized to the surrounding neuropil in ImageJ 1.44 p software (NIH, USA). Cells with normalized fluorescence intensity >1.5 x neuropil staining (a level apparent in ~50% of all unfilled cells) were scored as SP + ve and thus D1 +ve.

For tissue analysis 2–3 and 12 month-old mice were terminally anesthetized (sodium penthobarbital 240 mg/Kg, i.p.) and perfused with 4% paraformaldehyde (PFA). Brains were removed, post-fixed (4% PFA 4°C) overnight then cryoprotected with sucrose. Coronal slices (30 μm) were obtained by cryostat and treated the next day with 10 mM sodium-citrated plus 0.05% Tween (30 min, 37°C). Sections were rinsed with 0.1% PBST (3 × 10 min), blocked in 3% milk in 0.03% PBST (30 min, RT) followed by a second block in 10% NGS in 0.03% PBST (30 min, RT). Primary polyclonal rabbit anti-VGluT1 (135302 Synaptic Systems; 1:1000), guinea pig anti-VGluT2 (AB2251 Millipore; 1:2500), chicken Tyrosine Hydroxylase (ab76442; Abcam; 1:1000) and affinity isolated rabbit anti-Dopamine Transporter (N-terminal, D6944 Sigma; 1:1000) antibodies were applied (5% NGS + 0.01% $NaN_3$ in PBST; overnight 4°C) prior to washing (3 × 10 min PBST) and secondary incubation with goat Alexa-fluor-488 anti-rabbit, Alexafluor-633 anti-guinea pig, Alexafluor-488 anti-chicken and Alexafluor-568 anti-rabbit IgG secondary antibodies (90 min RT, Invitrogen; 1:1000), washed (PBST 3 × 10 min) and mounted with DAPI Fluoromount-G (Catalog: 0100–20; SouthernBiotech). Images were acquired at 20x magnification on an Olympus Fluoview FV-1000 confocal laser scanning microscope. For gluta-matergic synapse stains (VGluT1 and VGluT2), 60x magnification z-sections (0.33 μm/slice) were flattened into five stack max projections and binarized to create puncta masks. Analysis of puncta area, integral intensity, density and colocalization were conducted using CellProfiler (pipelines available on request). For DAT and TH staining, fluorescent intensity measurements of the striatum and cortex were averaged for each section from five ROIs placed on 20X z-projected image stacks (1.0 μm/slice) using ImageJ software (NIH, USA). Background fluorescence intensities were subtracted from 5 ROIs acquired in the corpus callosum.

## Golgi silver stain impregnation

Animals were perfused with PBS, the brain removed and Golgi impregnated following manufacturer's instructions (FD NeuroTechnologies - Rapid GolgiStain Kit). SPNs in the dorsolateral striatum were identified in 150 μm coronal vibratome sections and imaged by light microscopy (10 and 40X, Zeiss Axio Observer.z1). 0.5 μm z-stacks were used to quantify dendritic protrusions in 3 x ~30 μm segments of tertiary and quaternary dendrites per cell (averages conducted per cell,~6 cells per animal) in imageJ, as in (*Milnerwood et al., 2013*; *Petkau et al., 2012*).

## Statistics and data reporting

Data are presented throughout as mean ± SEM where *n* is the number of animals, or else cells (electrophysiology)/brain slices (voltammetry/VGluT staining) from the number of mice indicated in parentheses. In *Figure 2—figure supplements 3*, *6* and *7*, t-test refers to an unpaired parametric Students' t-test. Multiple comparisons were conducted throughout by 2-way ANOVA or RM-ANOVA as indicated, and Holm-Sidak multiple comparison *post-hoc* tests were performed using Prism 6.0 (GraphPad, San Diego California USA).

# Additional information

## Funding

| Funder | Author |
| --- | --- |
| Michael J. Fox Foundation for Parkinson's Research | Matthew J Farrer<br>Austen J Milnerwood |
| Parkinson Canada | Mattia Volta<br>Stefano Cataldi<br>Chelsie A Kadgien<br>Austen J Milnerwood |
| Canadian Institutes of Health Research | Sarah E MacIsaac<br>Igor Tatarnikov |

Canada Excellence Research            Matthew J Farrer
Chairs, Government of Canada

The funders had no role in study design, data collection and
interpretation, or the decision to submit the work for publication.

## Author contributions

Mattia Volta, Dayne A Beccano-Kelly, Conceptualization, Data curation, Formal analysis, Investigation, Methodology, Writing—original draft, Writing—review and editing; Sarah A Paschall, Conceptualization, Data curation, Formal analysis, Investigation, Writing—original draft; Stefano Cataldi, Conceptualization, Data curation, Formal analysis, Investigation, Writing—original draft, Writing—review and editing; Sarah E MacIsaac, Formal analysis, Investigation, Writing—original draft, Writing—review and editing; Naila Kuhlmann, Data curation, Formal analysis, Investigation, Methodology, Writing—review and editing; Chelsie A Kadgien, Igor Tatarnikov, Jaskaran Khinda, Sabrina Bergeron, Data curation, Formal analysis, Investigation, Writing—original draft, Writing—review and editing; Jesse Fox, Emma Mitchell, Data curation, Formal analysis, Investigation; Heather Melrose, Resources, Writing—original draft, Writing—review and editing; Matthew J Farrer, Conceptualization, Resources, Supervision, Funding acquisition, Writing—original draft, Writing—review and editing; Austen J Milnerwood, Conceptualization, Resources, Data curation, Formal analysis, Supervision, Funding acquisition, Validation, Investigation, Visualization, Methodology, Writing—original draft, Project administration, Writing—review and editing

## Author ORCIDs

Mattia Volta (iD) https://orcid.org/0000-0002-0300-6796
Dayne A Beccano-Kelly (iD) https://orcid.org/0000-0003-3592-8354
Sarah A Paschall (iD) https://orcid.org/0000-0003-1440-4412
Stefano Cataldi (iD) https://orcid.org/0000-0001-7708-4124
Austen J Milnerwood (iD) https://orcid.org/0000-0002-0056-1778

## Ethics

Animal experimentation: Mice were maintained according to Canadian Council on Animal Care regulations and the University of British Columbia Animal Ethics Committee (UBC AAC certification A16-0088 & A15-0105)

## Decision letter and Author response

Decision letter https://doi.org/10.7554/eLife.28377.015
Author response https://doi.org/10.7554/eLife.28377.016

## Additional files

### Supplementary files

• Transparent reporting form
DOI: https://doi.org/10.7554/eLife.28377.014

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
