## [Decision Letter]

[Editors’ note: a previous version of this study was rejected after peer review, but the authors submitted for reconsideration. The first decision letter after peer review is shown below.]

Thank you for submitting your work entitled "Progressive behavioral and synaptic phenotypes in LRRK2 G2019S knock-in mice" for consideration by *eLife*. Your article has been evaluated by a Senior Editor and four reviewers, one of whom, Dr. Andrew West, is a member of our Board of Reviewing Editors. The three peer reviewers have opted to remain anonymous.

Our decision has been reached after consultation between the reviewers and editors that all agree on the recommendation.

All reviewers thought there were many strong and exciting points to the work. The construct validity is particularly impressive, as was the analysis at multiple time points. However, as explained below, all reviewers raised issues revolving around three main points. Unfortunately the nature and severity of the points raised does not align to a reasonable path for acceptance of your work at *eLife* in the short-term, and should therefore lead to a rejection notice so that the work would not be further delayed.

There are three major points in summary that preclude further consideration that we hope will be helpful:

1) There are major problems with the statistics employed. All reviewers noted the appearance of arbitrarily conceived tests and group comparisons. In many cases the type of test being used did not seem justified, for example a post-hoc test within the context of a non-significant ANOVA, and the tendency to interpret a non-significant result as a trend. Because a thorough re-evaluation of the tests being used may dramatically alter the conclusions that would be drawn in a revision, this is the central problem that should preclude further consideration. *eLife* weighs very seriously concerns about statistics.

2) The behavior analyses are needed to support the conclusions but reviewers felt the data that were included were insufficient and may be over-interpreted to fit into the model, and substantial new experiments would be needed to convince the field of this aspect of the study.

3) The last point is that reviewers objected to the muddled style of the manuscript, where novel and confirmatory points are blended together in a broad-stroke tendency to over-generalize the results in a way reminiscent of an opinion piece or editorial. All reviewers felt the arguments made were thought provoking and well-conceived, but out of place in this context. With broad arguments made but little data to convincingly support them, especially in the electrophysiology, this opens the door for the need for a slew of new experimental approaches that are needed in order for the conclusions to be supported in a way that convinces the reviewers.

*Reviewer #1:*

The manuscript by Volta et al. describes a follow-up to their original characterization of the LRRK2 G2019S knockin (GKI) mouse model of Huntington's disease (HD). The authors use an array of behavioral, electrophysiological, electrochemical and anatomical approaches to characterize young (1-3 month old) and middle-aged (12 month old) mice. The approaches are all ones with which the authors have demonstrated expertise. The authors conclude that there is an age-dependent effect of the LRRK2 mutation. In young mice, it leads to hyperactivity and increased striatal glutamatergic and dopaminergic transmission. In the older mice, these effects reverse. However, the data doesn't provide convincing support for either set of conclusions (as stated).

There are other concerns. The most robust behavioral effect is the reduction in open field and rearing behavior in the older GKI mice. The conclusion that the behavioral effects in the GKI mice are 'biphasic' rests up a modest increase in rearing in the cylinder test in young GKI mice. This modest effect could have several causes and might not be related to 'increased exploratory drive' or dopamine at all. The analysis of presynaptic glutamatergic function is muddled. Moreover, the data do not support the conclusion that release probability is elevated at glutamatergic synapses on SPNs in young GKI mice. First, spontaneous EPSC frequency in ex vivo brain slices is complicated by a range of factors, including slice angle, which limits its information value. Second, the absence of an alteration in evoked PPR in the GKI mice argues that there is not a change in release probability, assuming that electrical stimulation and sEPSCs are giving a picture of the same synapses. As a consequence, the conclusion that glutamatergic synaptic transmission is enhanced in young GKI mice is not justified.

The slowing of the sEPSCs with age in the GKI SPNs is one of the cleanest pieces of data in the paper (Figure 2). But why is there the change? If this is dependent upon presynaptic spiking and not upon release probability (no change in PPR or apparent synapse density), it becomes very difficult to interpret.

Although interesting, the effects of quinpirole are very difficult to interpret because there are D2Rs not just at presynaptic sites but postsynaptic ones as well, opening the door to all sorts of indirect neuromodulatory effects (glutamatergic terminals have a wide range of GPCRs capable of altering release). Were the effects of quinpirole on evoked glutamate release dependent upon SPN subtype? What effect did a D2R antagonist have on evoked properties? It is also worth noting that these experiments are complicated by the use of intrastriatal stimulation (which activates a wide range of chemically distinct axons). Either cortical stimulation in parasagittal slices or optogenetic activation of cortical axons would have simplified this analysis.

The proposition that DA release increased in GKI striata with low frequency stimulation because of 'altered vesicle cycling and availability' is interesting but without any experimental support.

*Reviewer #2:*

This is an interesting and well-crafted study examining the age-dependent effects of the LRRK2 G2019S mutation in mice on motor behavior and medium spiny neuron activity. The strengths of the study sit with the longitudinal analysis of the mice, the identification of mutation and age-dependent cellular and behavioral phenotypes and with the characterization of changes in dopamine availability. These strengths are diluted by the failure to test the mechanism analyzed (DA availability) as a source of the changes in behavior or glutamatergic activity. Novel and confirmatory results are presented in a way that is unnecessarily confusing and some of the statistics used are inappropriate.

1) The data show the LRRK2 G2019S point mutation can affect motor behavior in a cylinder and spontaneous EPSCs in SPNs (Figure 1, Figure 2). Subsequent data (Figure 2, 4, 5) support that there are genotype dependent differences in dopamine "tone" in the dorsal striatum. Since the dopamine alterations reported could cause the altered motor behavior and influence cellular behavior, it is not clear why this mechanism was not tested directly at any age. Additionally, would LRRK2 kinase inhibitors be anticipated to "normalize" behavior and activity in GKI mice?

2) Data presentation gives equal space to novel and confirmatory findings. This is confusing and detracts from what is novel. Confirmatory findings should be presented more briefly and used to establish rationale and/or demonstrate reproducibility. Novel data require more thorough analyses. For example novel longitudinal studies should be presented in a manner that permits direct comparison between genotypes and ages. Data shown in Figure 1Ci, ii; 1Di, ii repeat published experiments by the same group. These data and new experiments (1Bi, ii) are each subjected to separate, pairwise comparisons. Aside from being statistically weak, this presentation style fails to fully assess effects of age and genotype (using something like a two-way ANOVA). Figure 4Aii and iii should also be reconfigured along these lines. Such direct comparisons would be more robust, feature what is novel, and could reveal new information.

3) Performance in the cylinder test is broadly interpreted as hyper- or hypoactivity. This is not a typical interpretation of this test. Differences in rearings alone are insufficient to conclude the mice are hyperactive in their youth and hypoactive in old age and this oversimplification needs to be reined in. (The slight decrease in "exploration" at 12 months in KI mice could support hypoactivity, but contradiction with previous work from this lab and others suggests that at best, this effect is weak). However, since rearing number reveals a genotype-dependent difference in young and old animals, the video data could (and should) be further analyzed. In addition to numbers of rearings, does duration change? Was there any evidence of forepaw dyskinesia (since forelimb wall contacts were measured)? Citation/s should also be provided for the method.

4) The idea that the G2019S mutation may be conferring early synaptic senescence is intriguing (Abstract, Discussion), but not well supported by the data. Changes observed in older WT neurons could reflect aging, but they do not occur earlier in GKI neurons (normal aging in both?); other changes in GKI neurons are not matched by WT neurons (GKI pathology?). The statement that the mouse model "recapitulates data from LRRK2 carriers" (Abstract) is a stretch.

5) Direct comparisons between two data sets using t-test (i.e. # and ## in Figure 2Aii, iii) that are part of another analysis (ANOVA with a post test) are not appropriate (subsection “Striatal glutamate transmission”, second paragraph; also Figure 4, Figure 5ii). If ANOVA shows no difference then a posthoc test cannot be used. Sample variance should also be taken into account and F-stats are needed. The same data is presented twice in Figure 2Cii, and then again in iii. If this is intentional, then proper statistical analysis should be done, comparing all three groups (WT+Q, GKI+Q, WT-Q), and groups should be shown on the same set of axes. Statistical test used for Figure 1 is not provided.

6) Changes in PPR support alterations in presynaptic terminal function, but are insufficient to conclude altered release probability as suggested.

7) Introduction and Discussion are overly long and there is a tendency to over-generalize. For example, "mediated by a multiplicity of interactions with Rab-family GTPases", has not yet been tested for the functions outlined, and "a progressive, biphasic, phenotype in GKI mice, reminiscent of that observed in heterozygous human carriers."-How are any of the phenotypes described progressive and what biphasic phenotype is being referred to in humans?

7) Why were electrophysiological recordings and FSCV conducted at RT? Temperature has effects on the measures recorded.

8) Figure 3 does not add to what has been published by this group and others and could be deleted.

9) There are also some errors in the Materials and methods describing LY fills and immunohistochemistry (LY and 488?). Where does Cy5 filling come from and what is examined in the Cy3 channel? Raster scanned image stacks are generally unreliable for intensity measures (Figure 5) – why not use a western blot for DAT (previous work in the lab shows TH is unchanged)?

10) Example traces should be added to Figure 1 for the 12 or 18 month data where genotype-dependent changes in EPSC frequencies are observed.

*Reviewer #3:*

Generally, I think the subject matter – modeling early, pre-degenerative changes in synaptic function in PD models is important and of brand interest.

I think the authors have been very thorough in their approach.

My concerns are mainly with the range of statistical analyses employed that appear rather random in places, and that some of the conclusions may be lacking a solid data-based foundation.

Specifics (as they arise in the manuscript rather than in any order of importance):

Figure 1: why is the n for the rearing experiments much smaller than the other 2 behavioral measures? Were they conducted in independent groups of animals? Or were outliers removed?

Subsection “Striatal glutamate transmission”, second paragraph: are the quoted ANOVA significance levels (0.001) referring to the interaction of age and genotype for both measures?

Subsection “Striatal glutamate transmission”, second paragraph: why Bonferroni for one comparison and Dunn for another? And Mann Whitney for another?

Figure 4: why are DA levels normalized to WT (presumably normalized to the average of the WT data)? Raw data not significant? (Not done for DOPAC+HVA:DA ratio).

Figure 4 and subsection “Striatal dopamine levels and release”, first paragraph: trend towards "altered" DA turnover (p=0.33) is barely a trend.

The microdialysis data might have been more informative had they conducted no-net-flux analysis to gain insight into DA clearance and provide comparison with the FSCV data.

Figure 4Biii and 5Aii and in associated text: what are the numbers in parentheses? Is this the number of animals? And the former number of stimulations? Is this legitimate?

Further explanation of the factor "tau" would be helpful. Unclear to me why the authors conclude increased release is the cause of the longer decay times in evoked transients in the mutant when the amplitude of these transients is not increased. (Although the amplitudes of the specific transients from which tau was calculated are not given and perhaps should be). I understand the fact that the difference in tau persists in the presence of the uptake blocker argues against impaired uptake being a factor (although I think this might be better explained), but perhaps differences in other forms of clearance (diffusion/extracellular volume fraction/tortuosity?) should be considered/discussed ahead of concluding it is facilitated release/recycling.

Having said that, the rise in transient amplitudes with repeated stimulations over time at low frequency is perhaps one of the most robust findings of the manuscript.

I'm also not sure the DAT immunofluorescence is a sufficiently sensitive measure of membrane-associated DAT, but kudos to them for considering "altered" expression as an explanation for reduced effectiveness of the uptake inhibitor. (But again, the rationale could be better explained).

To further conclude that the changes are due to vesicular recycling may be considered a bit of a stretch based on these data alone, it seems to me, although of course there are plenty of other reasons to think this is likely.

[Editors’ note: what now follows is the decision letter after the authors submitted for further consideration.]

Thank you for submitting your article "Age-related decline of behavioral alterations, glutamate and dopamine neurotransmission in LRRK2 G2019S knock-in mice" for consideration by *eLife*. Your article has been favorably evaluated by a Senior Editor and four reviewers, one of whom, Andrew West, is a member of our Board of Reviewing Editors.

The reviewers have discussed the issues with one another and the Reviewing Editor has drafted this decision to help you prepare a revised submission.

The principle finding of the manuscript is that young mice have elevated striatal glutamate and dopamine release and show increased dopamine extracellular lifetime compared to WT littermate control mice. Background literature surrounding LRRK2 would support that some of these outcomes would be related to increased LRRK2 activity due to enhanced levels of LRRK2 and/or enhanced kinase activity. The high frequency of the mutation in PD makes these first reports of this novel model (freely available commercially) of high interest.

However, the report is largely descriptive with few new mechanisms proposed, and the report is very short on behavioral outcomes.

For further consideration, all reviewers felt that the interpretation that increased sEPSCs (in this study) are due to increased mEPSCs should be tested directly by recording in the presence of TTX. Just as important, a D2R antagonist should be applied to attempt to ameliorate the knock-in effects. These experiments in the 'young' mice will be required to bolster the impact to a minimum level for further consideration.

Should you choose to resubmit with the additional data requested above, there are several minor adjustments (text only, no new data) that are needed to clarify some points:

1) For all voltammetry slice experiments, how many slices from how many animals for each group needs to be made clearer in the legend of Figure 3. The reviewers are assuming a well-powered experiment. If the data points (and stats) are number of slices from a much smaller number of animals, this would be unconventional in the voltammetry literature.

2) Representative western blots should be presented for all analyses (e.g., western blots for DAT), and all densitometry data (LiCOR, etc.) should be included as a supplementary excel file that shows individual calls. For example, readers could then be able to tell, for example, whether there might be GADPH differences, and how good of a 'housekeeper' the protein is, based on the non-normalized data which at the moment is hidden.

3) The term "age-related decline" and similar phrases used show bias in the outcomes compared to more neutral terms like "normalize" that should be considered.

4) The text still rambles in the introduction, for example the discussion of non-motor symptoms in PD. Focusing in on the GS mutation and past e-phys studies may be more useful to readers. The authors may need additional assistance from others to improve readability and the storyline in stitching together the three seemingly disparate main datasets together in a more convincing way.

5) Data shown in Figure 2—figure supplement 2 suggests that the < 6 month group was < 3 month for this dataset. If true, the narrower range should be reported, as the changes reported may not be sustained at 6 months.

---

## [Author Response]

[Editors’ note: the author responses to the first round of peer review follow.]

All reviewers thought there were many strong and exciting points to the work. The construct validity is particularly impressive, as was the analysis at multiple time points. However, as explained below, all reviewers raised issues revolving around three main points. Unfortunately the nature and severity of the points raised does not align to a reasonable path for acceptance of your work at eLife in the short-term, and should therefore lead to a rejection notice so that the work would not be further delayed.There are three major points in summary that preclude further consideration that we hope will be helpful:1) There are major problems with the statistics employed. All reviewers noted the appearance of arbitrarily conceived tests and group comparisons. In many cases the type of test being used did not seem justified, for example a post-hoc test within the context of a non-significant ANOVA, and the tendency to interpret a non-significant result as a trend. Because a thorough re-evaluation of the tests being used may dramatically alter the conclusions that would be drawn in a revision, this is the central problem that should preclude further consideration. eLife weighs very seriously concerns about statistics.

Regarding the statistics, this is only a matter of presentation, and there were no arbitrary selections. Specifically, non-parametric tests were chosen based on statistical testing of normal distribution (explained in Materials and methods), and the only reason we separated the behaviour by age was to highlight that it was not the same animals over time, obviously that is the case for the terminal experiments, so we didn't.

We have simplified the presentation and analyses to 2-way ANOVA throughout, with HolmSidak post-test. While many (legitimately analysed) statistically significant differences may be masked by limiting analyses to one post-test throughout, in many cases this maintains the same statistical significance pattern. Thus for clarity we have blanket analysed all data. At no point was a direct comparison reported on grouped data with a failed (p>0.05) ANOVA, merely as a less stringent post-test on a successful ANOVA (which was clearly explained). I urge the editors and reviewers to note that p-values estimate how unlikely it is that data sets are different by chance alone (depending on how the p value is generated, i.e., which post-test is used and its alpha adjustment), not whether or not they are ‘really different’.

2) The behavior analyses are needed to support the conclusions but reviewers felt the data that were included were insufficient and may be over-interpreted to fit into the model, and substantial new experiments would be needed to convince the field of this aspect of the study.

We have similarly simplified the presentation and discussion of the behavioural results.

3) The last point is that reviewers objected to the muddled style of the manuscript, where novel and confirmatory points are blended together in a broad-stroke tendency to over-generalize the results in a way reminiscent of an opinion piece or editorial. All reviewers felt the arguments made were thought provoking and well-conceived, but out of place in this context. With broad arguments made but little data to convincingly support them, especially in the electrophysiology, this opens the door for the need for a slew of new experimental approaches that are needed in order for the conclusions to be supported in a way that convinces the reviewers.

We feel that what has been referred to as confirmatory points are in several places important controls. Other than the microdialysis (which we have removed as previously published in data from the founding colony), we have presented many of these data here as supplemental. The arguments regarding what the data might mean (which I would call discussion, not conclusions) have been removed or, we hope, more explicitly discursive than conclusive.

Reviewer #1:The manuscript by Volta et al. describes a follow-up to their original characterization of the LRRK2 G2019S knockin (GKI) mouse model of Huntington's disease (HD).

The G2019S mutation is liked to Parkinson’s disease, not Huntington’s disease. We assume this is just an elaborate typo and request that the reviewer be more careful in future appraisals.

The authors use an array of behavioral, electrophysiological, electrochemical and anatomical approaches to characterize young (1-3 month old) and middle-aged (12 month old) mice. The approaches are all ones with which the authors have demonstrated expertise. The authors conclude that there is an age-dependent effect of the LRRK2 mutation. In young mice, it leads to hyperactivity and increased striatal glutamatergic and dopaminergic transmission. In the older mice, these effects reverse. However, the data doesn't provide convincing support for either set of conclusions (as stated).

For clarity, and at the reviewer’s suggestion, we have presented the data throughout as ‘young’ and ‘aged’, to match other data sets within the manuscript, and justified this with regards to our previous study in the text. We have toned-down the argument and discussion of the implications but maintain that the data forces a conclusion that [the mutation]: “In young mice, it leads to hyperactivity and increased striatal glutamatergic and dopaminergic transmission”. It would appear that older animals did/do not have robust exploratory differences; in the current manuscript we do not discuss a reversal of phenotype, but age-dependent changes.

There are other concerns. The most robust behavioral effect is the reduction in open field and rearing behavior in the older GKI mice. The conclusion that the behavioral effects in the GKI mice are 'biphasic' rests up a modest increase in rearing in the cylinder test in young GKI mice.

There was no statistical evidence that the later-aged observations in open field and rearing were the most robust finding in the previous document (far from it); to be frank, a n of 9-12 is low for behavioural observations. Thanks to the reviewers, and revision process, we have had the opportunity to increase observations of 12+ month-of-age animals, as presented in this short report. Having increased our observations and power, the reduction in open field exploration is not apparent and cylinder rearing remains as nothing more than a trend in old animals. We are very grateful for the opportunity to have shored up the data set, and are relieved that (unbiased, statistical) over-interpretation of a low n has been avoided.

We do not find the conclusions rest upon a ‘modest’ effect in young animals. The 25% increase in exploratory rearing is robust, as highlighted by the comparison of a high number of observations in all young animals (<6 months). In this submission, we discuss the data only in terms of significant genotype, age or age-interaction effects.

This modest effect could have several causes and might not be related to 'increased exploratory drive' or dopamine at all.

We did not endeavor to claim dopamine alterations had a causal effect, but that they were temporally correlated. In this manuscript we have attempted to limit speculation and discussion has been restricted to the minimum necessary to report the findings with context.

The analysis of presynaptic glutamatergic function is muddled.

The data presentation and text has been massively simplified and several conclusions clearly discussed.

Moreover, the data do not support the conclusion that release probability is elevated at glutamatergic synapses on SPNs in young GKI mice.

Having eliminated several possibilities, and found that the PPRs were different in the presence of quiniprole, we concluded that elevated PR was the most plausible explanation (see agreement from reviewer 3). However, in the current manuscript, we have endeavored to ensure it is clear that we ‘discuss the possibilities’ without endorsing specific conclusions.

First, spontaneous EPSC frequency in ex vivo brain slices is complicated by a range of factors, including slice angle, which limits its information value. Second, the absence of an alteration in evoked PPR in the GKI mice argues that there is not a change in release probability, assuming that electrical stimulation and sEPSCs are giving a picture of the same synapses.

We agree, and (previously and currently) discussed the findings in this light. We address all these factors openly in the manuscript. With respect to slice angle, we have previously been concerned about this, but found no effect of TTX on sEPSCs in our hands (See Milnerwood 2010, Supplement as detailed in text here).

As a consequence, the conclusion that glutamatergic synaptic transmission is enhanced in young GKI mice is not justified.

Regardless of the underlying cause (increased Pr, differences between miniature or evoked release, pathway specificity), glutamatergic transmission is clearly enhanced in young GKI mouse brain slices. This is not a subtle alteration, given a 47% increase in mean event frequency, as such I see no reason why it is unacceptable to conclude that increases are apparent. We hope that in the revised document (with discussion less robustly highlighted) the reviewer appreciates this.

The slowing of the sEPSCs with age in the GKI SPNs is one of the cleanest pieces of data in the paper (Figure 2). But why is there the change? If this is dependent upon presynaptic spiking and not upon release probability (no change in PPR or apparent synapse density), it becomes very difficult to interpret.

It would seem a major stumbling block for this reviewer may be that this study only partially replicates the results of a recent J Neuroscience paper by the Huntley and Benson labs (Matikainen-Ankney 2016). We discuss this in the document, and the Matikainen-Ankney et al.study contains some data that is very similar to our own at 1 month, but didn't show an increase in older slices (2 months, n not disclosed). There are several issues that preclude direct comparisons (e.g., homo vs. het comparison, WT background vs. littermates); however, while there was not a significant increase in mEPSC frequency (leading to a conclusion that actionpotential firing underlies all difference) in the Matikainen-Ankey study, this may be due to reduced power in highly variable data e.g., Matikainen-Ankey detected a strong trend towards increased mEPSC frequency (Figure 4.A*p*=0.09, in a cell sample size of 16&20 neurons). Our observation numbers were higher in isolated 1 month data (n=24&36) and combined <6 month data (n=40&53).

Although interesting, the effects of quinpirole are very difficult to interpret because there are D2Rs not just at presynaptic sites but postsynaptic ones as well, opening the door to all sorts of indirect neuromodulatory effects (glutamatergic terminals have a wide range of GPCRs capable of altering release). Were the effects of quinpirole on evoked glutamate release dependent upon SPN subtype? What effect did a D2R antagonist have on evoked properties?

As we discussed previously, and in this submission of the document, there are several confounding factors to take into account when attempting to interpret the D2 agonism result; that said, it is unclear why this negates the ‘interesting’ nature of the observation. We found that Pr was different between two genotypes in the presence of the drug. We also explain how this is likely due to an effect on WT, rather than GKI, as statistically described. We have included more data in Figure 2—figure supplement 5 to show there were no obvious differences in event frequency of D1 and D2 expressing SPNs in GKI mice. As requested we have included data that show there was no differential effect of quinpirole on evoked amplitudes, nor PPRs, in D1 or D2 GKI SPNs. We feel this data goes someway to assuage concerns that post-synaptic cell type, or differential post-synaptic effects of quinpirole, might negate the conclusion that glutamate transmission is (transiently) elevated in GKI brains.

It is also worth noting that these experiments are complicated by the use of intrastriatal stimulation (which activates a wide range of chemically distinct axons). Either cortical stimulation in parasagittal slices or optogenetic activation of cortical axons would have simplified this analysis.

We do not shy from this fact, and this was noted in the original document. The reviewer complains that the study is too long and complicated, while demanding more ‘confirmatory’ experiments (on only half of the document’s contents). We maintain that the results are of interest, and agree that optogenetic activation and / or different slice orientations are entirely appropriate follow-up studies; but ones that are outside of the scope of the current manuscript, especially in a short report format.

The proposition that DA release increased in GKI striata with low frequency stimulation because of 'altered vesicle cycling and availability' is interesting but without any experimental support.

We have removed reference to altered vesicle cycling and availability.

Reviewer #2:This is an interesting and well-crafted study examining the age-dependent effects of the LRRK2 G2019S mutation in mice on motor behavior and medium spiny neuron activity. The strengths of the study sit with the longitudinal analysis of the mice, the identification of mutation and age-dependent cellular and behavioral phenotypes and with the characterization of changes in dopamine availability.

We thank the reviewer for their appraisal of the study’s general strengths. However, there has been some confusion regarding the use of the term ‘longitudinal’ (see reviewer 3), which suggests that the same animals were tested over time. This was obviously not the case for terminal experiments, but was also not the case for behavioural tests, in which the repeated exposure of the animals to testing might alter responses.

These strengths are diluted by the failure to test the mechanism analyzed (DA availability) as a source of the changes in behavior or glutamatergic activity.

We appreciate it would be ideal to test the link between dopamine availability, glutamate activity and behavior, but stress that this is already a fairly ambitious study (the financial cost of raising 12-18 month old mice alone is extremely high). We hope that condensing the manuscript down to a short report will help the reviewer conclude that the findings merit publication ‘as is’, with follow-up investigations to probe the mechanism for some of these novel observations.

Novel and confirmatory results are presented in a way that is unnecessarily confusing and some of the statistics used are inappropriate.

We have attempted to remove ‘confirmatory’ experiments, but consider some valuable for interpretation of the novel observations. This and statistical concerns will be addressed below.

1) The data show the LRRK2 G2019S point mutation can affect motor behavior in a cylinder and spontaneous EPSCs in SPNs (Figure 1, Figure 2). Subsequent data (Figures 2, 4, 5) support that there are genotype dependent differences in dopamine "tone" in the dorsal striatum. Since the dopamine alterations reported could cause the altered motor behavior and influence cellular behavior, it is not clear why this mechanism was not tested directly at any age. Additionally, would LRRK2 kinase inhibitors be anticipated to "normalize" behavior and activity in GKI mice?

We assume the reviewer wishes we conduct whole-animal pharmacology and test for alterations to behavior and neuronal physiology; that this should be attempted for dopamine modulators and LRRK2 kinase inhibitors. Firstly, now that we have a defined early phenotype in GKI mice it is plausible to conduct pharmacological experiments at the early time point. This is a very reasonable suggestion, but is a matter of breeding large cohorts of animals (especially for littermate comparisons) for use over several months. We do intend to do such experiments, and feel while they will add greatly to our programmatic aims; however, we feel they are outside the mandate for this manuscript, especially as a short report. For the assessment of LRRK2 kinase inhibitor experiments to be conducted this is even more so. While the third generation of LRRK2 inhibitors appear to be fairly selective, we would certainly need to include LRRK2 knock-out (and littermates) mice in comparison to the GKI mice (and littermates) in treated and untreated groups, over time. Many animals would need to be drugged independently of the actual experiment to assay resulting CNS levels and target engagement. Furthermore, there is the issue of when to drug animals; if drugs were only given to mice at phenotypic time points it may be informative if the hypothesis is proven, but if not it might produce a false negative. In other words, if differences are due to altered development, inhibitors would need to be given from a very early time point. We accept that these experiments should be conducted, but this will require many tens (even hundreds of mice) and it is perhaps better to publish the phenotypes we have already uncovered for a push by the whole field (including industry) to test their ‘druggability’. This is an important but not a small addition to our programmatic aims, but we feel this level of pharmacological investigation would require an entirely separate manuscript.

2) Data presentation gives equal space to novel and confirmatory findings. This is confusing and detracts from what is novel. Confirmatory findings should be presented more briefly and used to establish rationale and/or demonstrate reproducibility. Novel data require more thorough analyses. For example novel longitudinal studies should be presented in a manner that permits direct comparison between genotypes and ages. Data shown in Figure 1Ci, ii; 1Di, ii repeat published experiments by the same group. These data and new experiments (1Bi, ii) are each subjected to separate, pairwise comparisons. Aside from being statistically weak, this presentation style fails to fully assess effects of age and genotype (using something like a two-way ANOVA). Figure 4Aii and iii should also be reconfigured along these lines. Such direct comparisons would be more robust, feature what is novel, and could reveal new information.

We have removed many of the ‘confirmatory’ data, repurposed some as supplemental data, and now have little in the way of discussion for them in the short report format. The statistical analyses have been standardized throughout as 2-way ANOVA

3) Performance in the cylinder test is broadly interpreted as hyper- or hypoactivity. This is not a typical interpretation of this test. Differences in rearings alone are insufficient to conclude the mice are hyperactive in their youth and hypoactive in old age and this oversimplification needs to be reined in.

We have altered the text accordingly.

(The slight decrease in "exploration" at 12 months in KI mice could support hypoactivity, but contradiction with previous work from this lab and others suggests that at best, this effect is weak). However, since rearing number reveals a genotype-dependent difference in young and old animals, the video data could (and should) be further analyzed. In addition to numbers of rearings, does duration change? Was there any evidence of forepaw dyskinesia (since forelimb wall contacts were measured)? Citation/s should also be provided for the method.

A much improved older data set demonstrates that hypoactivity is not apparent in older mice. We could further analyse the cylinder test videos for further information at the reviewers’ suggestion in the short report; however, we are wary of over interpreting the data in the cylinder test (rearing itself is fairly unambiguous and simple to score consistently). We did also quantify grooming time and grooming bouts, but found no alterations. We have included recent citations for our own previous use of these tests, to which we could add, but this seems unnecessary. We do thank the reviewer for the suggestion and as we embark on future pharmacological investigations we will endeavor to take this advice into account e.g., forelimb dyskinesia has been induced by repeated L-DOPA treatment of mitoPark (but not control) mice (Gellhaar et al. 2015).

4) The idea that the G2019S mutation may be conferring early synaptic senescence is intriguing (Abstract, Discussion), but not well supported by the data. Changes observed in older WT neurons could reflect aging, but they do not occur earlier in GKI neurons (normal aging in both?); other changes in GKI neurons are not matched by WT neurons (GKI pathology?).

We have removed this phrasing from the document. We retain the observation for early changes in GKI that are matched by WT for the increase in dopamine extracellular lifetime.

The statement that the mouse model "recapitulates data from LRRK2 carriers" (Abstract) is a stretch.

Agreed, this has been amended to a more general comparison between the slow progression of phenotypic changes in the mouse model with that in human LRRK2 patients

5) Direct comparisons between two data sets using t-test (i.e. # and ## in Figure 2Aii, iii) that are part of another analysis (ANOVA with a post test) are not appropriate (subsection “Striatal glutamate transmission”, second paragraph; also Figure 4, Figure 5ii). If ANOVA shows no difference then a posthoc test cannot be used. Sample variance should also be taken into account and F-stats are needed. The same data is presented twice in Figure 2Cii, and then again in iii. If this is intentional, then proper statistical analysis should be done, comparing all three groups (WT+Q, GKI+Q, WT-Q), and groups should be shown on the same set of axes. Statistical test used for Figure 1 is not provided.

At no point was a direct comparison (two groups, t-test) performed on grouped data (> two groups) for which there was not a significant ANOVA. Statistical analysis and presentation have been standardised throughout with F values included.

6) Changes in PPR support alterations in presynaptic terminal function, but are insufficient to conclude altered release probability as suggested.

Interpretation has been simplified, and other potential contributions discussed.

7) Introduction and Discussion are overly long and there is a tendency to over-generalize. For example, "mediated by a multiplicity of interactions with Rab-family GTPases", has not yet been tested for the functions outlined, and "a progressive, biphasic, phenotype in GKI mice, reminiscent of that observed in heterozygous human carriers."-How are any of the phenotypes described progressive and what biphasic phenotype is being referred to in humans?

Introduction and Discussion has been simplified throughout in this short report.

8) Why were electrophysiological recordings and FSCV conducted at RT? Temperature has effects on the measures recorded.

While it is true that recordings are influenced by temperature (among many other variables), recording at room temperature is standard for assessment of basal physiological properties in slice preparations; conducting experiments at temperatures below 37^o^C prevents the build-up of condensation on electrodes which can cause stimulation and recording points to change over long time periods. Regardless, all conditions were constrained for all mice, slices, experiments and with the investigator blinded to genotype (the independent variable); we make no claims on absolute biophysical properties at physiological temperature.

9) Figure 3 does not add to what has been published by this group and others and could be deleted.

Demonstrating that synapse numbers are unaltered is critical to interpretation of the data. No other group has assessed these measures in these mice, although similar findings have been reported in related models. With that in mind, these data are retained as supplemental.

10) There are also some errors in the Materials and methods describing LY fills and immunohistochemistry (LY and 488?). Where does Cy5 filling come from and what is examined in the Cy3 channel? Raster scanned image stacks are generally unreliable for intensity measures (Figure 5) – why not use a western blot for DAT (previous work in the lab shows TH is unchanged)?

We apologise for confusion. A section of the Materials and methods contained elements of a previous Materials and methods section with some aspects incorrect for this manuscript. Cells were filled with Lucifer Yellow, or biocytin. Biocytin was visulalized by Cy3-conjugated streptavidin. I am unsure why the reviewer feels that the image acquisition is not reliable for DAT intensity (the stacks were constrained in every way, fluorescence signal is linear, and ‘background’ was accounted for relative to signal on the corpus callosum). We have additionally included semi quantitative western blots in the figure supplement, which demonstrate an increase in DAT (as shown very recently in similar KI mice by Longo et al. 2017). Regardless, the slowing of transients occurred in a very high concentration of DAT blocker, eliminating this as a possible cause of increased DA extracellular lifetime.

11) Example traces should be added to Figure 1 for the 12 or 18 month data where genotype-dependent changes in EPSC frequencies are observed.

Genotype-dependent changes in EPSC frequency were not observed in the older group.

Reviewer #3:[…] My concerns are mainly with the range of statistical analyses employed that appear rather random in places, and that some of the conclusions may be lacking a solid data-based foundation.Specifics (as they arise in the manuscript rather than in any order of importance):Figure 1: why is the n for the rearing experiments much smaller than the other 2 behavioral measures? Were they conducted in independent groups of animals? Or were outliers removed?

No outliers were removed from any data sets. In one group of animals, video data for one cylinder test group were corrupted. Added to this, on occasion mice of either genotype leap clear of the beaker and it is not found fair to include them in rearing analyses.

Subsection “Striatal glutamate transmission”, second paragraph: are the quoted ANOVA significance levels (0.001) referring to the interaction of age and genotype for both measures?

Throughout the (current) manuscript, we have explicitly reported ANOVA main and interaction effects (with associated F values) and post test analyses.

Subsection “Striatal glutamate transmission”, second paragraph: why Bonferroni for one comparison and Dunn for another? And Mann Whitney for another?

Generally statistical post-tests were determined by the nature of the data spread (regarding normal distributions). Otherwise, we were acting with knowledge of post-hoc testing being a sliding scale of significance. All post hoc analyses adjust for alpha in different ways and some are more stringent than others e.g., Bonferroni will make fewer Type 1 errors by testing each comparison by a harsh alpha adjustment. This has the unfortunate effect of being highly prone to false negatives due to the harsh alpha adjustment. Thus, which ANOVA post test is used informs the magnitude of confidence that type 1 errors are avoided, not whether data are ‘truly different or not’. To avoid confusion, we have standardized statistical analyses of similarly presented data throughout.

Figure 4: why are DA levels normalized to WT (presumably normalized to the average of the WT data)? Raw data not significant? (Not done for DOPAC+HVA:DA ratio).Figure 4 and subsection “Striatal dopamine levels and release”, first paragraph: trend towards "altered" DA turnover (p=0.33) is barely a trend.The microdialysis data might have been more informative had they conducted no-net-flux analysis to gain insight into DA clearance and provide comparison with the FSCV data.

DA levels were normalized to account for differences in HPLC runs, potentially between standards, but also with regards to length of storage prior to assay. There was no need to normalize ratio’s as they are internally controlled (within sample). In retrospect, should we conduct microdialysis in future, no-net flux technique will be employed. That said, we conducted the experiment to satisfy our own need to replicate the reduced DA levels found in the initial colony at Mayo, in the UBC colony and we have removed microdialysis data from the manuscript in the short report format.

Figure 4Biii and 5Aii and in associated text: what are the numbers in parentheses? Is this the number of animals? And the former number of stimulations? Is this legitimate?

As stated in the Materials and methods n=cells or slices for voltammetry. The number of animals from which these cells/slices were prepared from is in parenthesis. This is standard practice.

Further explanation of the factor "tau" would be helpful. Unclear to me why the authors conclude increased release is the cause of the longer decay times in evoked transients in the mutant when the amplitude of these transients is not increased. (Although the amplitudes of the specific transients from which tau was calculated are not given and perhaps should be). I understand the fact that the difference in tau persists in the presence of the uptake blocker argues against impaired uptake being a factor (although I think this might be better explained), but perhaps differences in other forms of clearance (diffusion/extracellular volume fraction/tortuosity?) should be considered/discussed ahead of concluding it is facilitated release/recycling.Having said that, the rise in transient amplitudes with repeated stimulations over time at low frequency is perhaps one of the most robust findings of the manuscript.

We hope that the revised document satisfies the reviewers concerns regarding the explanation for increased extracellular lifetime / transient tau, and the correct references.

I'm also not sure the DAT immunofluorescence is a sufficiently sensitive measure of membrane-associated DAT, but kudos to them for considering "altered" expression as an explanation for reduced effectiveness of the uptake inhibitor. (But again, the rationale could be better explained).

In addition to the immunofluorescence we have also included semi quantitative western blots in the figure supplement, which demonstrate an increase in DAT (as shown very recently in similar KI mice by Longo et al. 2017). It is unlikely that the slowing of transients occurred due to a deficit in DAT function as we used a very high concentration of DAT blocker, eliminating this as a possible cause of increased DA extracellular lifetime. We hope the reviewer finds the revised document better explained.

To further conclude that the changes are due to vesicular recycling may be considered a bit of a stretch based on these data alone, it seems to me, although of course there are plenty of other reasons to think this is likely.

We also believe there are plenty of reasons to consider this likely; however, due to concerns raised we have eliminated reference to vesicle recycling. We feel the document may suffer from a paucity of sensible speculation / discussion and welcome further suggestions in this regard.

[Editors' note: the author responses to the re-review follow.]

[…] For further consideration, all reviewers felt that the interpretation that increased sEPSCs (in this study) are due to increased mEPSCs should be tested directly by recording in the presence of TTX. Just as important, a D2R antagonist should be applied to attempt to ameliorate the knock-in effects. These experiments in the 'young' mice will be required to bolster the impact to a minimum level for further consideration.

Both of these experiments have been conducted and included as two supplemental figures. While strengthening the manuscript slightly, the conclusions remain unaltered.

Should you choose to resubmit with the additional data requested above, there are several minor adjustments (text only, no new data) that are needed to clarify some points:1) For all voltammetry slice experiments, how many slices from how many animals for each group needs to be made clearer in the legend of Figure 3. The reviewers are assuming a well-powered experiment. If the data points (and stats) are number of slices from a much smaller number of animals, this would be unconventional in the voltammetry literature.

This has been addressed.

2) Representative western blots should be presented for all analyses (e.g., western blots for DAT), and all densitometry data (LiCOR, etc.) should be included as a supplementary excel file that shows individual calls. For example, readers could then be able to tell, for example, whether there might be GADPH differences, and how good of a 'housekeeper' the protein is, based on the non-normalized data which at the moment is hidden.

This has been addressed: representative blots are shown and an. xls document detailing the densitometry analysis is included.

3) The term "age-related decline" and similar phrases used show bias in the outcomes compared to more neutral terms like "normalize" that should be considered.

We feel at no point are phrases such as ‘age-related decline’ misused, and reflect the changes in the data. It doesn’t appear often, but does so as part of discussion and conclusion. Opinion in this regard is certainly a matter for debate, but we include no descriptive inaccuracies.

4) The text still rambles in the introduction, for example the discussion of non-motor symptoms in PD. Focusing in on the GS mutation and past e-phys studies may be more useful to readers. The authors may need additional assistance from others to improve readability and the storyline in stitching together the three seemingly disparate main datasets together in a more convincing way.

We have reduced and simplified certain aspects of the introduction, but feel the message remains valid. We had non-experts read this text and have not received reports of difficulty with the narrative. We feel the document is clearly written, and while there may be some expectation for the audience to consider for themselves how the data sets are linked, this is done deliberately in order to maintain impartiality and avoid sensationalised ‘story-telling’, which plagues modern scientific publication.

5) Data shown in Figure 2—figure supplement 2 suggests that the < 6 month group was < 3 month for this dataset. If true, the narrower range should be reported, as the changes reported may not be sustained at 6 months.

Age ranges have been more specifically reported as 1-3, 2 or 3 months within each data set.